engineering geology/environmental engineering

iron ore mine, transition from open-pit to underground mining, water inrush characteristics, hazard effects, hydrochemical analysis

**Author for correspondence:**
Bin Zhang
e-mail: sc_zhb@cugb.edu.cn

# Water inrush characteristics and hazard effects during the transition from open-pit to underground mining: a case study

Huijie Zhang, Bin Zhang, Nengxiong Xu, Lei Shi, Hanxun Wang, Weiru Lin and Yiwei Ye

School of Engineering and Technology, China University of Geosciences (Beijing), Beijing 100083, People's Republic of China

(ID) BZ, 0000-0003-3452-1413

During the transition from open-pit to underground mining in iron ore mines, water inrush is a prominent problem for mine safety and production. In this paper, a comprehensive method that incorporates hydrochemical analysis and numerical simulation is proposed to analyse the characteristics of water inrush during the transition from open-pit to underground mining. The proposed method revealed the migration law of groundwater and analysed the source of mine water inrush in the Yanqianshan iron mine located in Liaoning province, China. The results show that the excavated mine roadway is the primary factor affecting groundwater migration and that the source of the mine water inrush is the groundwater in the aquifer around the mine roadway. Moreover, based on the results of the study, appropriate methods for prevention and treatment of mine water inrush were proposed. This approach provides a novel idea for the assessment of water inrush hazards and will serve as a valuable reference for analogous engineering cases.

## 1. Introduction

Iron ore, being one of the most important industrial raw materials, is exploited and consumed in large quantities in many industries, and its usage is increasing every year [1,2]. The open-pit mining method is generally preferred in mining activities due to its extensive applicability to various kinds of exposed rock, its minimum production loss and its high production rate, especially for materials such as iron ore, bauxite and copper ore, which are

**Figure 1.** Geographical location of the study area.

exposed at the surface [3]. Moreover, the transition from open-pit to underground mining is often required during mining operations to maximize resource exploitation and increase economic efficiency, as some deposits extend from the shallow surface to a great depth [4,5], such as the Chah-Gaz iron mine in Iran [6], Udachny mine in Russia [7] and Shirengou iron mine in China [8].

In general, mine exploitation inevitably changes the natural geological and structural environment and leads to the development of exposed water conductive faults, watery karst caves and water-filled gobs, which may cause mine water inrush [9]. Existing studies have focused on optimizing the transition from open-pit to underground mining [10,11]. However, as mining depths increase and mining faces expand, more serious water inrush problems can occur during the transition from open-pit to underground mining.

In recent years, research on the mine water inrush problem has drawn increasing attention. Empirical coefficients and mathematical methods have been applied to predict mine water inrush [12–15], while physical experiments, numerical simulations and field measurements have been applied to study the mechanisms of mine water inrush (groundwater seepage, flow-stress-damage, etc.) [16–18]. In order to provide a scientific basis for water inrush prevention in mining districts, it is of vital importance to determine the cause and source of water inrush. Therefore, the correct and effective identification of the source of mine water inrush is an urgent problem [19–22]. Because of its simplicity, efficiency and low cost, hydrochemical analysis is widely used in the identification of water sources [23–25]. Hydrochemical analysis originated from simple water quality analysis [26,27] and gradually developed into mathematical statistical methods. Ding *et al*. [28] used multivariate statistical methods to study heavy metals in the gold mine soil of the upstream area of a metropolitan drinking water source. According to Pourjabbar *et al*. [29], the characteristics of pore water and slate samples were critically analysed using fuzzy hierarchical cross-clustering statistical techniques to investigate the source of contamination near an abandoned uranium mine in Germany. However, research methods based on traditional statistical learning theory require a large number of samples and cannot meticulously describe the complex law of groundwater migration.

In this paper, a comprehensive method incorporating hydrochemical analysis and numerical simulation was proposed to analyse the characteristics of water inrush during the transition from open-pit to underground mining. The hydrochemical characteristics of water samples from the study area were analysed. Through the Schukalev classification method (SCM) and Piper diagrams, the hydraulic connection of groundwater in the study area was revealed. A three-dimensional groundwater seepage model was built to analyse the characteristics of mine water inrush. With the use of MODPATH particle inverse tracking, the migration law of groundwater was analysed and the source of water inrush at the mine roadway was identified. Based on this analysis, prevention and treatment measures were proposed to solve the problem of mine water inrush.

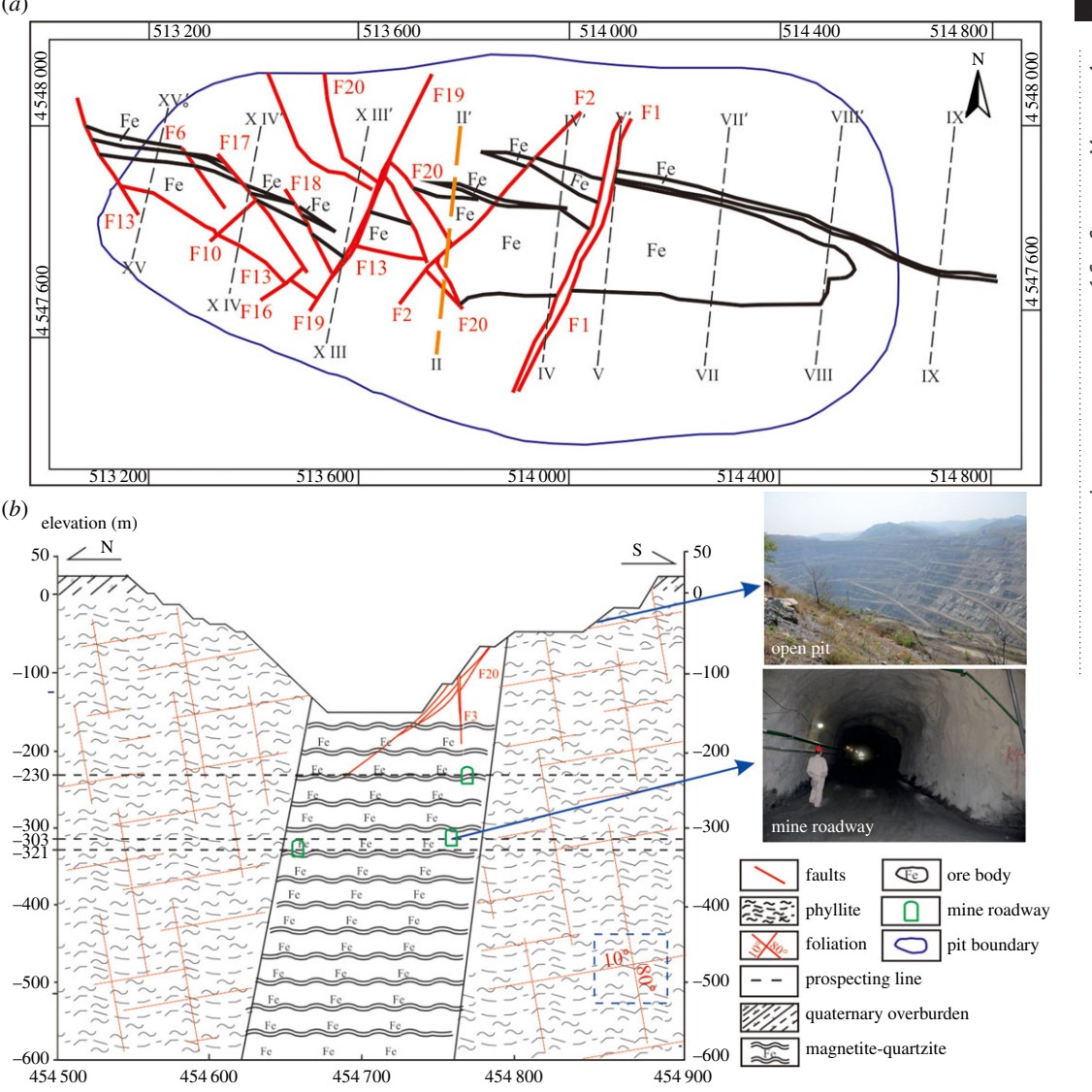

**Figure 2.** Geological condition of the study area. (*a*) Geological structure of mine area. (*b*) Geological section of II−II′ section.

# 2. Material and methods

## 2.1. Site description

### 2.1.1. Geological setting

The Yanqianshan iron mine is a large-scale open-pit mine located in Anshan city, Liaoning Province, China (figure 1). The landform in the study area is low mountain hilly land. The study area is surrounded by mountains with an elevation of 210–386 m to the south, east and north, while the area to the west consists of plains with an average elevation of approximately 93 m. The topography is high in the southeast and low in the northwest. The length of the pit is approximately 1410 m (from east to west), and the maximum width is approximately 710 m (from south to north). The elevation of the closed loop of the pit is 93 m [4]. The difference in elevation between the ultimate bottom and the closed loop of the pit is 270 m. To exploit the deep-seated ore bodies, the mining method transitioned from open-pit mining to underground mining, and some mine roadways were excavated.

The outcropped strata in the study area include the Archaean Anshan group, Lower Proterozoic Liaohe group and Cenozoic Erathem Quaternary overburden. The Anshan metamorphic rocks, including phyllite and magnetite-quartzite, are widely distributed over the study area, primarily along the northwest–southeast directions. Distribution of strata is phyllite, magnetite-quartzite and phyllite from north to south. The phyllite mainly contains two sets of dominant foliations and both sets of

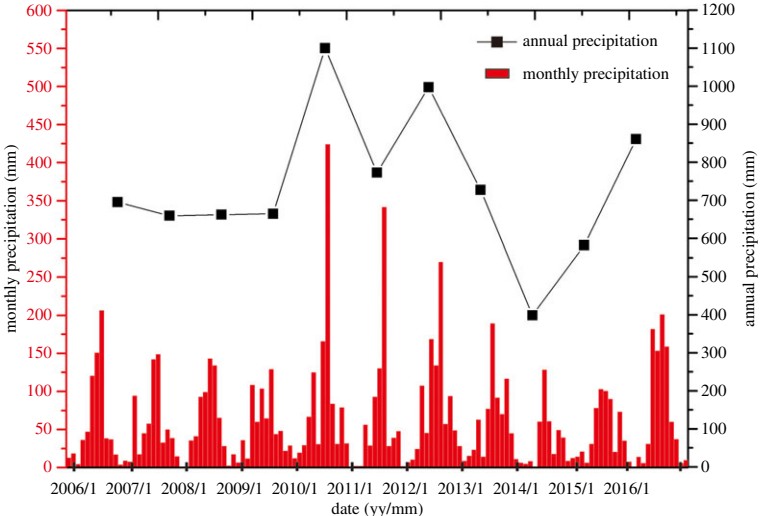

**Figure 3.** Precipitation of the study area during 2006–2016.

them were closed before excavation. The dip angles are $10°$ and $80°$, respectively (figure 2b). In this area, the iron ore band in magnetite-quartzite dips to the northeast at approximately $70–88°$. Gneissic mixed rocks, granite-like mixed rocks and chlorite schist are also developed in the study area. The ore body is cut by four main faults. F13 is normal fault with a vertical displacement of 160 m, which occurs in the northeast direction and whose dip angle is approximately $67–90°$. F20 is strike-slip fault with a horizontal displacement of 70 m, which occurs in the northeast direction and whose dip angle is approximately $45°$. F1 and F19 are strike-slip faults, with horizontal displacements of 40–130 m and 30–100 m, respectively, which occur in the southeast direction and whose dip angles are approximately $74°$ and $83–90°$, respectively (figure 2a).

### 2.1.2. Hydrogeology

The study area possesses a monsoon climate of medium latitudes. The annual average temperature is 9.6°C. July and August are the hottest months with an average temperature of 24.5°C, while January is the coldest month with an average temperature of $-8.6$°C. Precipitation is relatively low and concentrated. The evaporation effect is strong, with an annual average evaporation of 500 mm m$^{-2}$. According to precipitation statistics from 2006 through 2016, provided by Liaoning Statistical Information Net [30,31] (figure 3), the average annual precipitation is 738.2 mm. In addition, the precipitation is concentrated across the months of June, July and August, during which time the total precipitation reaches 421.31 mm and accounts for 57.07% of the annual precipitation.

The catchment area of the study area reaches $24 \times 10^6$ m$^2$ (figure 4). Due to its lower terrain, the mine pit acts as a convergence point for surface water. The surface water in the study area comprises the Guyu River, which is a seasonal river located to the south of the mine pit and which runs east to west. Recharge sources for the Guyu River include precipitation and Quaternary pore water. The main aquifers in the study area are Quaternary overburden and magnetite-quartzite (figure 2b). The Quaternary overburden is a phreatic aquifer, composed mainly of mild clay, sandy soil, sand gravel and gravel, 2.87–12.91 m thick with good water-richness. The groundwater is mainly recharged by precipitation. The discharge of the groundwater is mainly by artificial exploitation and outflow. The magnetite-quartzite is the bedrock fissure aquifer, 55–195 m thick, nearly upright distribution, cut by multiple faults. Recharge sources are precipitation and deep tectonic fissure water. The phyllite is the aquitard, to some extent limiting the groundwater flow in the aquifer. The faults (F1 and F19) have large water inflow in some mine roadways, and the hydraulic conductivity is strong.

### 2.2. Methods

Aiming to describe a comprehensive study of the characteristics of water inrush during the process of transition from open-pit to underground mining, a comprehensive method incorporating hydrochemical analysis and numerical simulation is proposed. Based on the aforementioned studies, prevention and

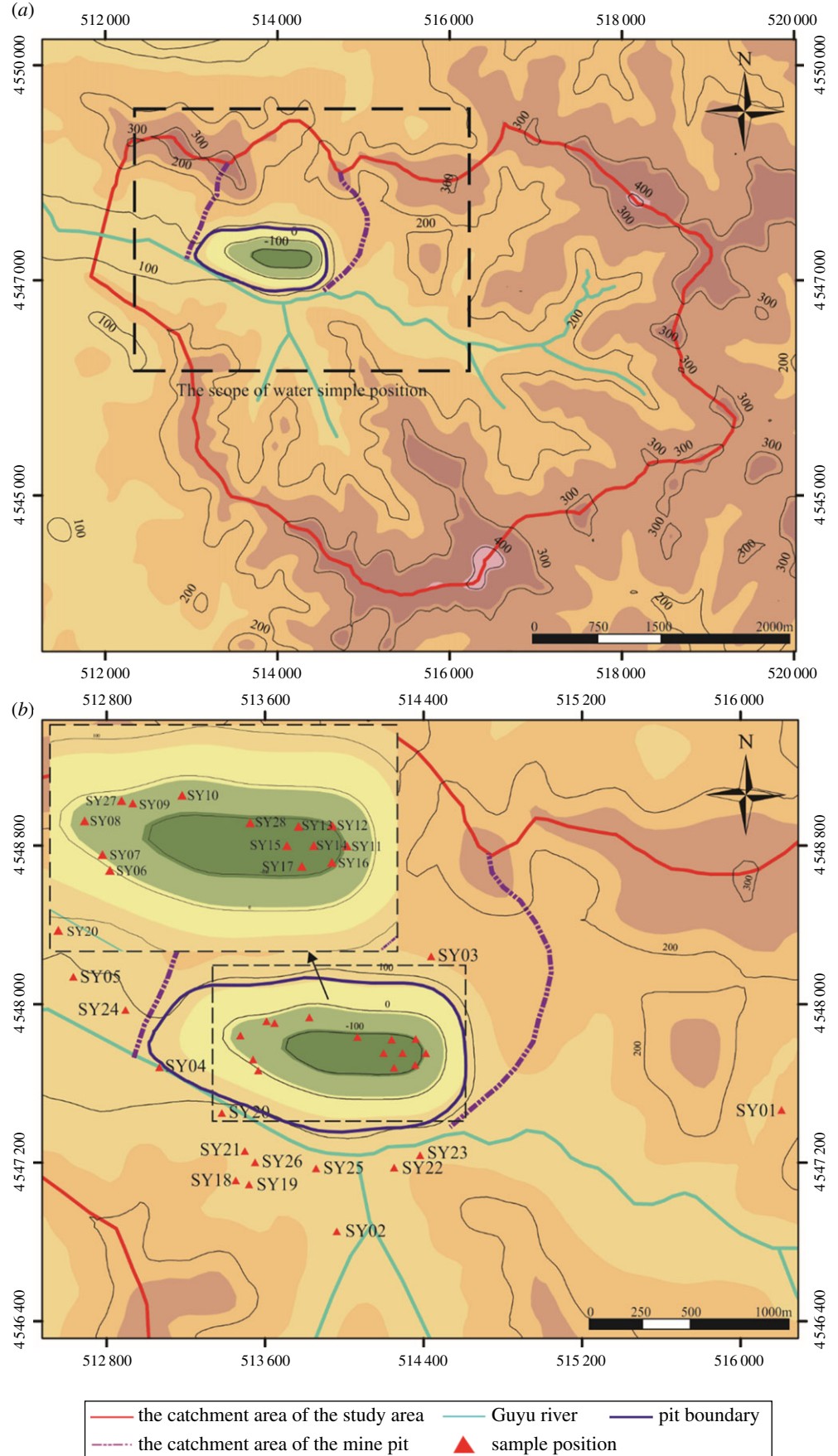

**Figure 4.** The distribution of surface water system in the study area. The distribution of (*a*) surface water in the study area and (*b*) water sample position.

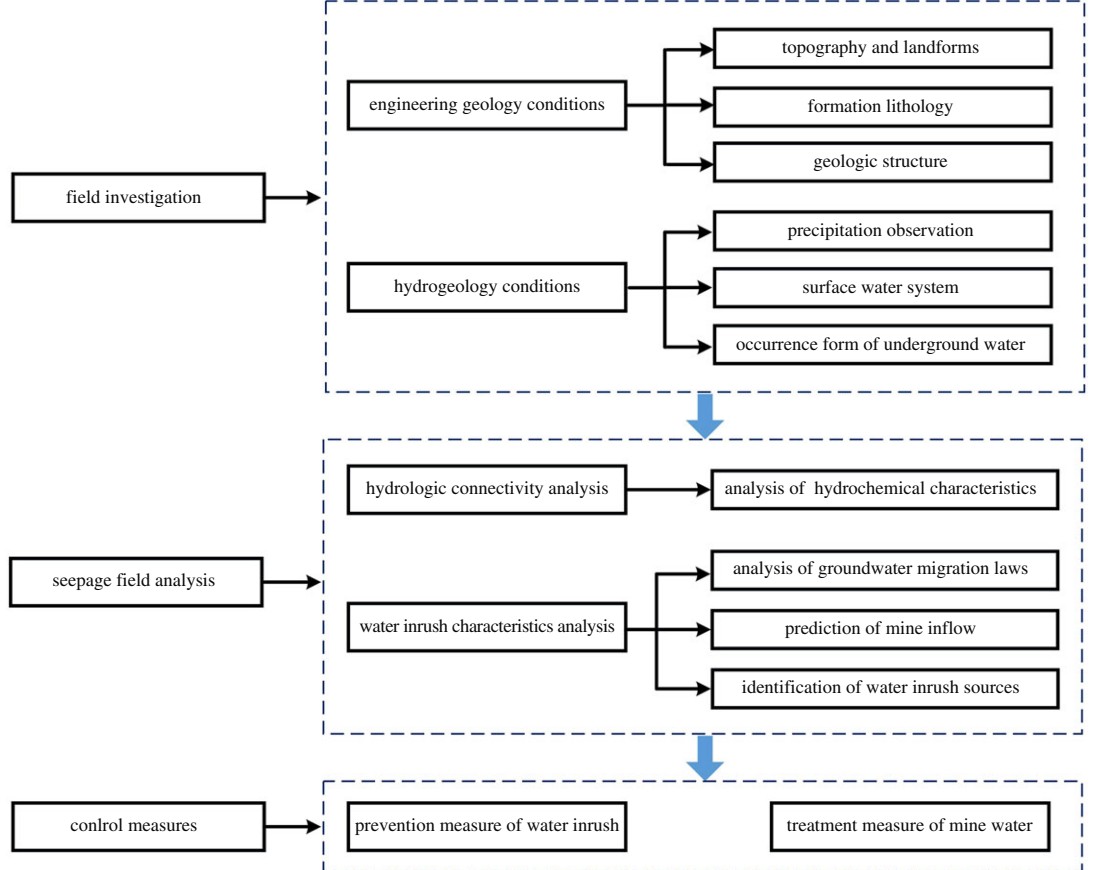

**Figure 5.** Flowchart of analyses conducted during this study.

**Table 1.** Hydrochemistry types classified by SCM.

| over 25% ion content | $HCO_3^-$ | $HCO_3^- + SO_4^{2-}$ | $HCO_3^- + SO_4^{2-} + Cl^-$ | $HCO_3^- + Cl^-$ | $SO_4^{2-}$ | $SO_4^{2-} + Cl^-$ | $Cl^-$ |
|---|---|---|---|---|---|---|---|
| $Ca^{2+}$ | 1 | 8 | 15 | 22 | 29 | 36 | 43 |
| $Ca^{2+} + Mg^{2+}$ | 2 | 9 | 16 | 23 | 30 | 37 | 44 |
| $Mg^{2+}$ | 3 | 10 | 17 | 24 | 31 | 38 | 45 |
| $Na^+ + Ca^{2+}$ | 4 | 11 | 18 | 25 | 32 | 39 | 46 |
| $Na^+ + Ca^{2+} + Mg^{2+}$ | 5 | 12 | 19 | 26 | 33 | 40 | 47 |
| $Na^+ + Mg^{2+}$ | 6 | 13 | 20 | 27 | 34 | 41 | 48 |
| $Na^+$ | 7 | 14 | 21 | 28 | 35 | 42 | 49 |

treatment measures were proposed to solve the problem of mine water inrush. The corresponding flowchart is shown in figure 5.

### 2.2.1. Sampling and hydrochemical analysis

Artificial exploitation significantly changes the concentrations of major ions in the groundwater, resulting in the complex and diverse hydrochemical characteristics in the mining area [32]. Hydrochemical classification is a comprehensive indicator for analysing the hydrochemical characteristics of groundwater. At present, the Shukalev classification method (SCM) is the most widely used in hydrochemical classification analysis, reflecting the law of groundwater migration [33]. The SCM is based on the main ion concentrations and mineralization in groundwater. Ion concentrations greater than 25 meq% are categorized into 49 types (table 1). The mineralization of water is categorized into four types: type A (less than 1.5 g l$^{-1}$), type B (1.5–10 g l$^{-1}$), type C (10–40 g l$^{-1}$) and type D (greater than 40 g l$^{-1}$). The two classifications—ion content and water mineralization—are combined into a

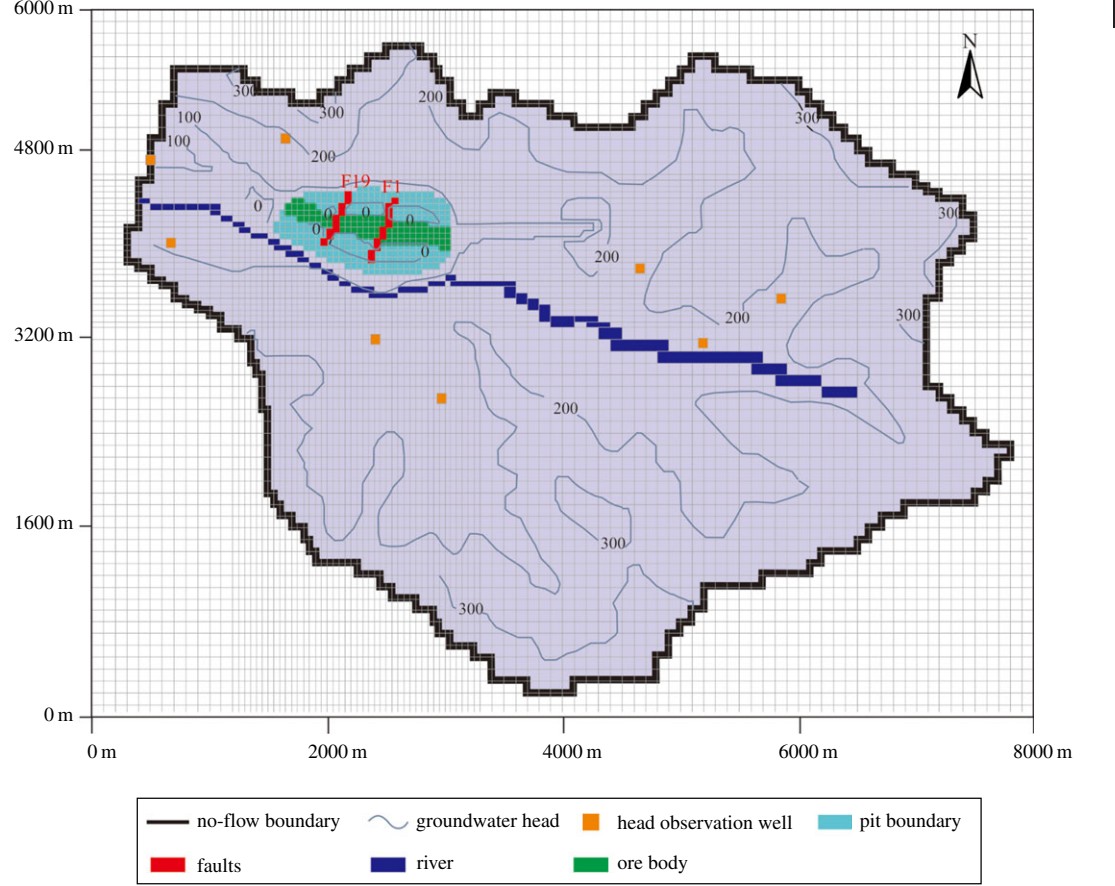

**Figure 6.** Model grid, boundary conditions and head observation well distribution.

designation of the form '1-A', for example, where 1 denotes the ion concentration and A denotes the water mineralization [34,35]. The hydrochemical characteristics of 28 water samples from the study area were analysed and studied (figure 4b). Samples SY01–SY05 are Quaternary pore water. Samples SY06–SY15 are bedrock fissure water. Samples SY16 and SY17 are goaf hydrops. Samples SY18–SY28 are mine drainage water. The concentrations of major ions, including $SO_4^{2-}$, $Cl^-$, $(Na^++K^+)$, $Ca^{2+}$ and $Mg^{2+}$, were estimated by ion chromatography (CIC-200). The alkalinity of major anions, including $HCO_3^-$ and $CO_3^{2-}$, was analysed by acid–base titration [36].

### 2.2.2. Groundwater flow modelling

#### 2.2.2.1. Theoretical model

Based on engineering geological and hydrogeological materials, a three-dimensional heterogeneous and anisotropic model was built to simulate the groundwater seepage field. The three-dimensional groundwater seepage equation was solved using a finite-difference method. The partial differential equation of groundwater seepage field used in the model is shown in the below equation

$$\frac{\partial}{\partial x}\left(K_{xx}\frac{\partial h}{\partial x}\right) + \frac{\partial}{\partial y}\left(K_{yy}\frac{\partial h}{\partial y}\right) + \frac{\partial}{\partial z}\left(K_{zz}\frac{\partial h}{\partial z}\right) - W = S_s\frac{\partial h}{\partial t} \tag{2.1}$$

where $K_x$, $K_{yy}$ and $K_{zz}$ are the hydraulic conductivity along the x-, y- and z-axes, respectively $(LT^{-1})$; h is the potentiometric head (L); W is the volumetric flux per unit volume, which represents sources or sinks of groundwater $(T^{-1})$; Ss is the specific storage $(L^{-1})$; and t is time $(T^{-1})$.

The solution of the governing partial differential equation satisfies the second type of boundary condition expressed by equation (2.2). The boundary of fixed water level is expressed by equation (2.3) [37]

$$K_n\frac{\partial h}{\partial \boldsymbol{n}}\bigg|_{\Gamma_2} = q(x, y, z, t) \quad x, y, z \in \Gamma_2, t \geq 0, \tag{2.2}$$

and

$$\nabla \cdot (\rho\bar{u}) = Q, \tag{2.3}$$

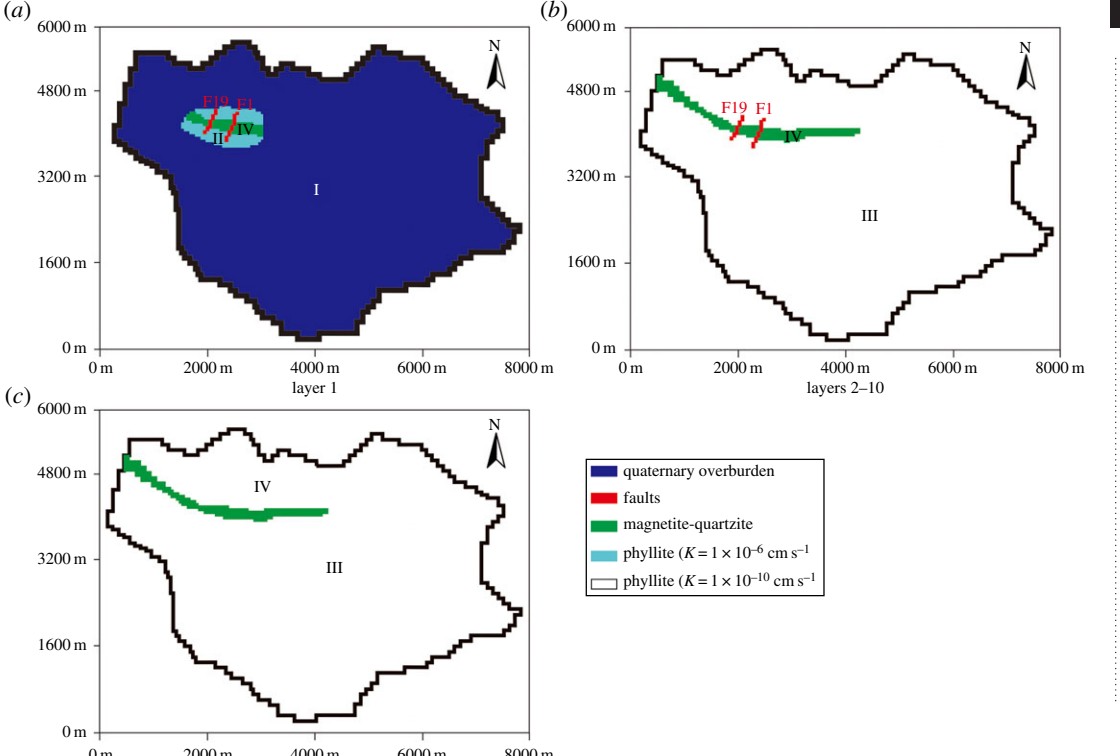

**Figure 7.** Permeability coefficient partitions in the study area.

**Table 2.** Hydrogeological parameters of the model.

|  | partition number | permeability coefficient $K$ (cm s$^{-1}$) | specific yield $S_s$ | specific storage $S_y$ | effective porosity | total porosity |
|---|---|---|---|---|---|---|
| quaternary | I | $8 \times 10^{-7}$ | 0.2 | $1 \times 10^{-5}$ | 0.15 | 0.3 |
| phyllite | II | $1 \times 10^{-6}$ | 0.008 | $1 \times 10^{-5}$ | 0.03 | 0.05 |
|  | III | $1 \times 10^{-10}$ | 0.0005 | $1 \times 10^{-5}$ | 0.015 | 0.02 |
| magnetite-quartzite | IV | $5 \times 10^{-5}$ | 0.0006 | $1 \times 10^{-5}$ | 0.02 | 0.025 |
| fault | F19 | $4 \times 10^{-6}$ | 0.15 | $1 \times 10^{-5}$ | 0.2 | 0.4 |
|  | F1 | $2 \times 10^{-6}$ | 0.1 | $1 \times 10^{-5}$ | 0.15 | 0.2 |

where $\rho$ is the density of fluid (kg m$^{-3}$); $q$ is the volume of water laterally flowing into or out of the aquifer per unit area and per unit time under the second type of boundary (m d$^{-1}$).

For three-dimensional steady flow, the mass balance equation of MODPATH can be expressed in terms of effective porosity and seepage velocity, which is shown in the below equation [38]

$$\frac{\partial(nV_x)}{\partial x} + \frac{\partial(nV_y)}{\partial y} + \frac{\partial(nV_z)}{\partial z} = W \tag{2.4}$$

where $V_x$, $V_y$ and $V_z$ are the components of the linear flow velocity vector along the $x$-, $y$- and $z$-axes, respectively (LT$^{-1}$); $n$ is the effective porosity of the aquifer (%); and $W$ is the volume of water provided by the sources or sinks per unit volume in aquifer (T$^{-1}$).

### 2.2.2.2. Numerical model

In addition to the engineering area, the model domain should be extended to the boundary of the hydrogeological unit to eliminate the uncertainty of boundary conditions of fractured rock mass. Therefore, the catchment area of the study area was chosen as the domain of the groundwater seepage model. The length of the three-dimensional groundwater seepage model was approximately

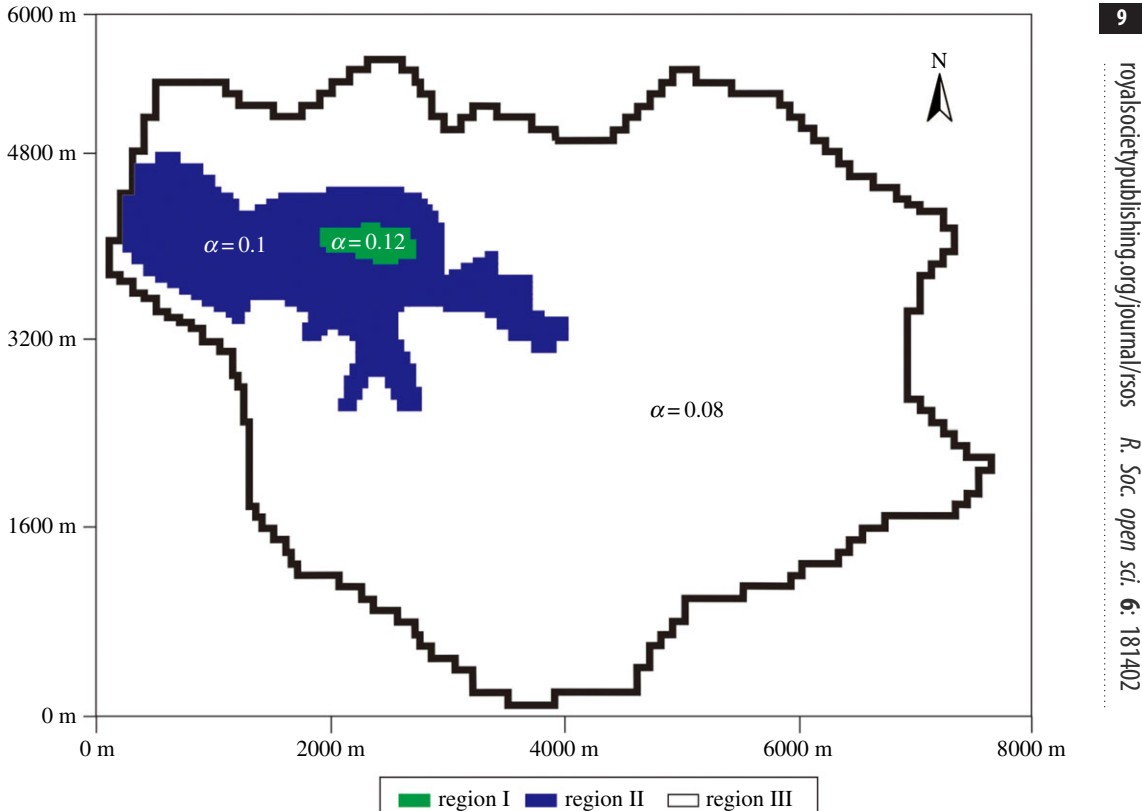

**Figure 8.** Infiltration coefficient partition in the study area.

8000 m (from east to west), and the width was approximately 6000 m (from south to north). The model consisted of a 100 × 100 m grid and a total of 88 476 cells. The grids near the mine pit were refined, and the grids outside the drainage divide were defined as inactive grids (figure 6). The model had 12 layers and the depth varied from −600 to 420 m. Layer 1 was an unconfined aquifer, and layers 2–12 included the confined aquifer and aquitard. Calculations were performed for unsteady flow with a time step of 30 days for a period of 60 months (January 2008–December 2012). After 5 years of calculation, the groundwater head change was basically stable, indicating that the period of time chosen to model is reasonable. According to the hydrogeological data, the hydraulic conductivity of the faults of F1 and F19 is strong, influencing the groundwater distribution in the study area. Therefore, only the influence of the faults of F1 and F19 on the groundwater seepage field was considered in the model. The sources of water input to the groundwater system are recharge from precipitation. The sources of water output from the groundwater system are groundwater evaporation and the Guyu River. The head observation wells were set in the model to observe the simulated values of the groundwater head, as shown in figure 6.

According to the monitoring data in the study area, the initial water head of the study area was obtained and assigned to the unsteady flow model. The domain of the groundwater seepage model was the catchment of the study area, bounded by the watershed. The groundwater on either side of the watershed does not flow to the other side; the volume of water at the watershed is zero. Therefore, the boundary condition of the model was set as the no flow in the second type of boundary condition. The boundaries of the river and mine roadway were set as the boundary of fixed water level (figures 6 and 10).

Obtained from previous studies and field investigation [39], the permeability coefficient partition used in the model is shown in figure 7, and the hydrogeological parameters for each region are shown in table 2. Region I was the phreatic aquifer, region IV was the bedrock fissure aquifer and regions II and III were the aquitard.

The amount of precipitation infiltration recharge in the study area is $5–10 \times 10^4 \, m^3$ per square kilometre, and the precipitation infiltration coefficient is approximately 0.06–0.13. Considering the precipitation characteristics, topographic features, soil (rock mass) permeability, vegetation characteristics, climatic conditions and evaporation in the study area, the infiltration coefficient partition was obtained (figure 8). The infiltration coefficient of region I, which mainly included the mine pit

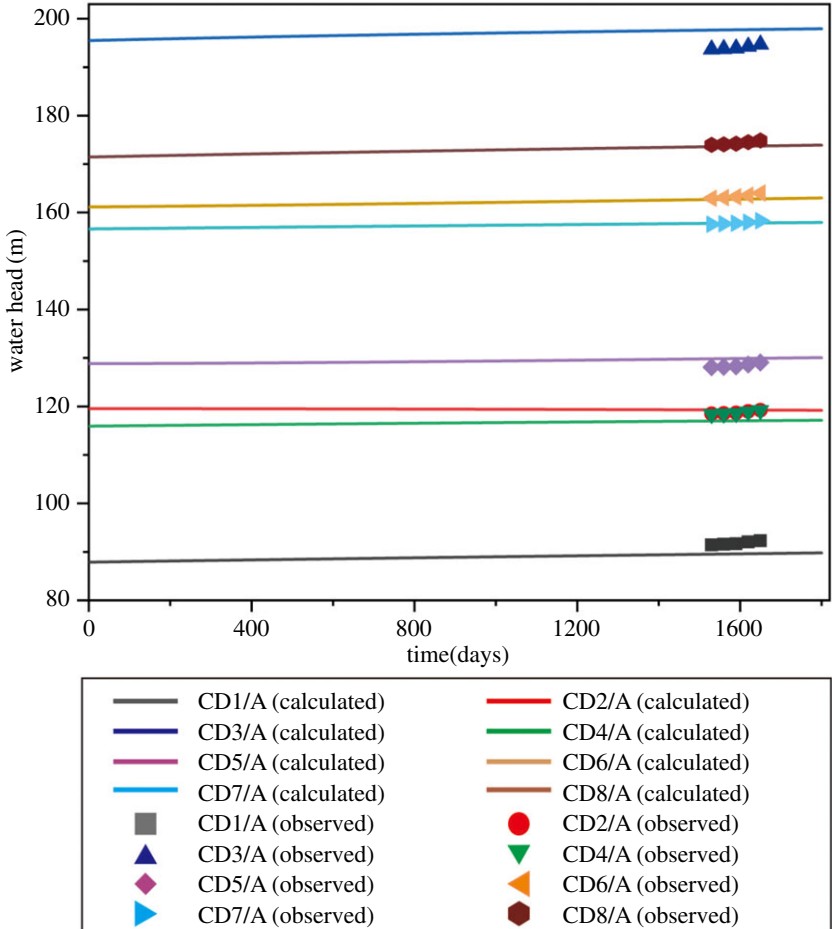

**Figure 9.** Comparison between the calculated head and observed head.

depression, was the highest. The infiltration coefficient of region II, which mainly included the steep slope around the pit, was higher than that of region I. The infiltration coefficient of region III, which mainly included gentle slopes and plains, was the lowest.

The model should be calibrated to ensure that it has the capacity to simulate the actual groundwater seepage field in the study area [40]. Based on the monitoring data, the numerical model was calibrated by adjusting the seepage parameters and boundary conditions. Comparing simulated values and measured values of the groundwater head, the capacity of simulating the actual groundwater seepage field in the study area can be evaluated. The results are shown in figure 9. The difference between the simulated values and the measured values was less than 1 m, and the confidence level was greater than 95%. The model calibration results prove that the three-dimensional groundwater seepage model effectively reflected the actual groundwater seepage field of the study area.

To exploit the deep-seated ore bodies, the mining method transitioned from open-pit mining to underground mining. The five horizontal mine roadways at elevations of −123, −213, −303, −321 and −501 m were excavated, as shown in figure 10. Changes in the groundwater head after roadway excavation were simulated in the calibrated model.

# 3. Results

## 3.1. Hydrologic connectivity analysis

The hydrochemical characteristics of water samples from the study area were analysed and studied via the SCM and Piper diagrams. The water samples were classified according to the SCM (table 3) [31]. The main anions in the water samples were $HCO_3^-$, $SO_4^{2-}$ and $Cl^-$, and the main cations were $Ca^{2+}$, $Na^+$ and $Mg^{2+}$. The ion concentrations of $CO_3^{2-}$ in samples SY22 and SY23 were $2.18 \, g \, l^{-1}$ and $15.24 \, g \, l^{-1}$, respectively, and their concentrations in the remaining samples were $0 \, g \, l^{-1}$. The pH values of the

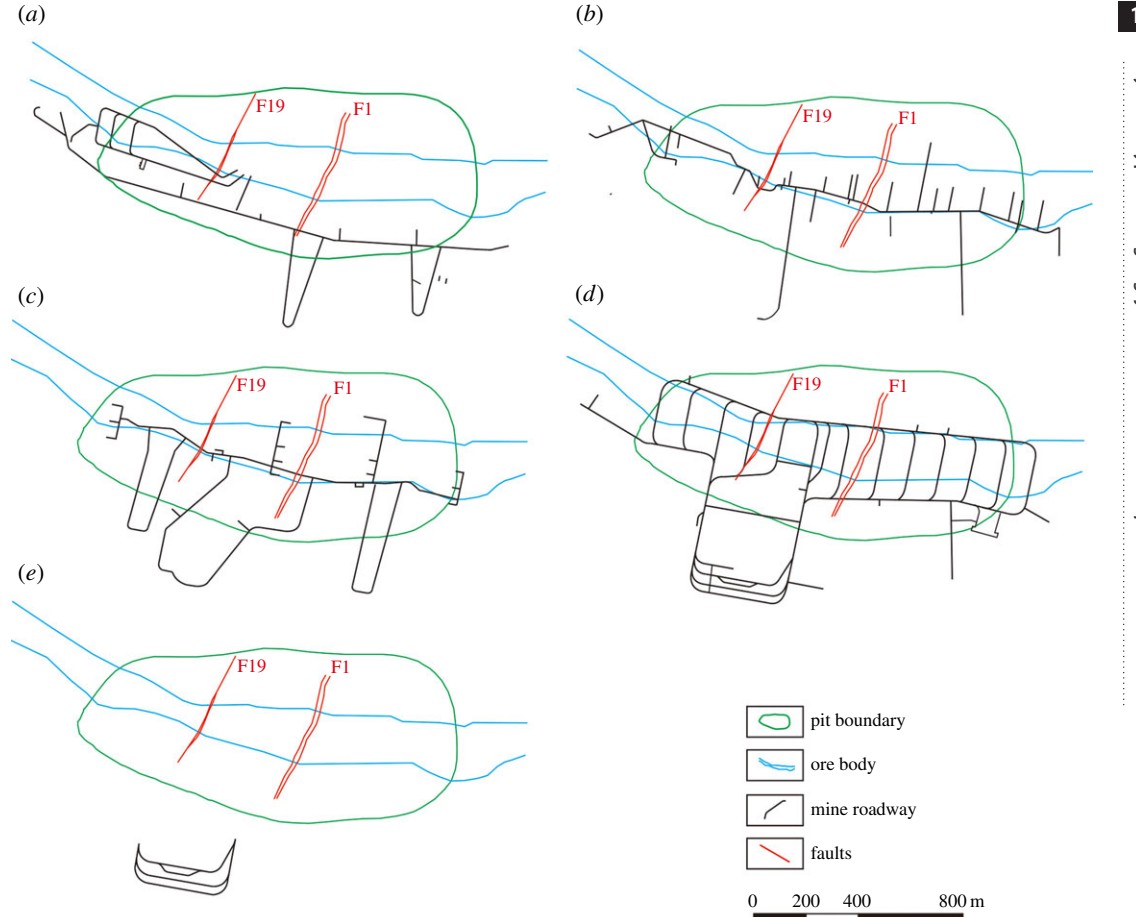

**Figure 10.** Layout of the mine roadways in the study area. (*a*) −123 m horizontal cross-section, (*b*) −213 m horizontal cross-section, (*c*) −303 m horizontal cross-section, (*d*) −321 m horizontal cross-section and (*e*) −501 m horizontal cross-section

samples were in the range of 6.75–8.93 with a mean of 7.86, and mineralization of the samples was essentially less than $1.5 \text{ g l}^{-1}$, indicating that the groundwater in the study area is generally neutral to weakly alkaline and that mineralization is low. Water samples from different aquifers possessed similar ion types and ion concentrations, indicating that similar hydrogeological and hydrodynamic conditions are prevalent among the aquifers. The percentages of $Mg^{2+}$ and $SO_4^{2-}$ were high in samples SY6, SY10, SY13 and SY14, indicating that key groundwater chemical components are significantly affected by the faults. Samples SY15, SY16 and SY17 are of a similar hydrochemical type, indicating that the goaf area is contaminated by mining activity.

A Piper diagram [41] was plotted to reveal the hydrologic connectivity between the different aquifers (figure 11). The water samples from the study area are basically located in zone 1, indicating good hydrologic connectivity between the aquifers. The quaternary pore water samples (SY04, SY05) above the iron body and most of the bedrock fissure water samples are located in zone 9; they have similar hydrogeochemical types, indicating that the source of the bedrock fissure water is quaternary pore water infiltrating along the fissures in the aquifer. In addition, the mine drainage water samples (SY18, SY19, SY21) located in the direction of horizontal extension of the excavated roadway have similar hydrogeochemical types, indicating that the excavated roadway provides a channel for groundwater in the aquifer and that the groundwater flows into the roadway and eventually is drained from the pit. The quaternary pore water samples (SY01, SY02) farther from the iron body and above the aquitard, as well as the mine drainage water samples (SY20, SY22, SY25, SY26), are located in zone 5, indicating that the source of water in the aquitard is phreatic water above the aquitard.

## 3.2. Water inrush characteristic analysis

The three-dimensional groundwater seepage model predicted that the water inflow in the mine roadways would be 81 016 $\text{m}^3\text{d}^{-1}$ after 5 years. As shown in figure 12, the groundwater head declined fastest in the

**Table 3.** Hydrochemical characteristics of water samples (unit, mg l$^{-1}$).

| sample type | sample | (K$^+$ + Na$^+$) | Ca$^{2+}$ | Mg$^{2+}$ | Cl$^-$ | SO$_4^{2-}$ | HCO$_3^-$ | pH | SCM (type) |
|---|---|---|---|---|---|---|---|---|---|
| quaternary pore water | SY01 | 30.62 | 66.65 | 16.29 | 21.32 | 88.18 | 202.6 | 7.04 | Ca$^{2+}$—HCO$_3^-$ + SO$_4^{2-}$ (8-A) |
| | SY02 | 12.83 | 17.91 | 9.05 | 9.84 | 35.73 | 74.26 | 6.75 | Na$^+$ + Ca$^{2+}$ + Mg$^{2+}$—HCO$_3^-$ + SO$_4^{2-}$ (12-A) |
| | SY03 | 41.35 | 200 | 80.25 | 20.95 | 638.9 | 162.1 | 7.67 | Ca$^{2+}$ + Mg$^{2+}$—SO$_4^{2-}$ (30-A) |
| | SY04 | 78.48 | 44.77 | 45.26 | 32.8 | 228.8 | 162.1 | 8.42 | Na$^+$ + Mg$^{2+}$—HCO$_3^-$ + SO$_4^{2-}$ (13-A) |
| | SY05 | 55.88 | 80.58 | 36.81 | 36.08 | 207.4 | 191.3 | 7.06 | Ca$^{2+}$ + Mg$^{2+}$—HCO$_3^-$ + SO$_4^{2-}$ (13-A) |
| bedrock fissure water | SY06 | 90.56 | 101.5 | 10.26 | 32.8 | 293.2 | 33.74 | 8.93 | Ca$^{2+}$—SO$_4^{2-}$ (29-A) |
| | SY07 | 68.36 | 93.51 | 13.88 | 32.8 | 214.6 | 189.1 | 8.10 | Ca$^{2+}$—HCO$_3^-$ + SO$_4^{2-}$ (8-A) |
| | SY08 | 61.56 | 119.4 | 18.1 | 26.24 | 250.3 | 222.8 | 7.95 | Ca$^{2+}$—HCO$_3^-$ + SO$_4^{2-}$ (8-A) |
| | SY09 | 62.02 | 133.2 | 12.07 | 29.53 | 238.4 | 222.8 | 8.06 | Ca$^{2+}$—HCO$_3^-$ + SO$_4^{2-}$ (8-A) |
| | SY10 | 68.2 | 64.66 | 35.6 | 41 | 209.8 | 195.9 | 8.13 | Ca$^{2+}$ + Mg$^{2+}$—HCO$_3^-$ + SO$_4^{2-}$ (9-A) |
| | SY11 | 52.64 | 42.78 | 32.58 | 22.96 | 102.5 | 189.1 | 7.64 | Ca$^{2+}$ + Mg$^{2+}$—HCO$_3^-$ + SO$_4^{2-}$ (9-A) |
| | SY12 | 43.23 | 70.63 | 45.26 | 26.24 | 221.7 | 202.6 | 7.67 | Ca$^{2+}$ + Mg$^{2+}$—HCO$_3^-$ + SO$_4^{2-}$ (9-A) |
| | SY13 | 298.8 | 112.4 | 82.67 | 36.08 | 633.4 | 168.8 | 7.75 | Ca$^{2+}$ + Mg$^{2+}$—SO$_4^{2-}$ (30-B) |
| | SY14 | 71.66 | 65.66 | 92.32 | 18.04 | 407.6 | 216.1 | 7.76 | Mg$^{2+}$—HCO$_3^-$ + SO$_4^{2-}$ (10-A) |
| | SY15 | 55.89 | 52.72 | 65.17 | 26.24 | 231.2 | 216.1 | 7.47 | Ca$^{2+}$ + Mg$^{2+}$—HCO$_3^-$ + SO$_4^{2-}$ (9-A) |
| goaf hydrops | SY16 | 40.26 | 59.69 | 60.34 | 24.6 | 264.6 | 175.5 | 7.42 | Ca$^{2+}$ + Mg$^{2+}$—HCO$_3^-$ + SO$_4^{2-}$ (9-A) |
| | SY17 | 41.92 | 83.56 | 59.74 | 21.32 | 317 | 162.1 | 7.22 | Ca$^{2+}$ + Mg$^{2+}$—HCO$_3^-$ + SO$_4^{2-}$ (9-A) |

(Continued.)

**Table 3.** (*Continued.*)

| sample type | sample | $(K^+ + Na^+)$ | $Ca^{2+}$ | $Mg^{2+}$ | $Cl^-$ | $SO_4^{2-}$ | $HCO_3^-$ | pH | SCM (type) |
|---|---|---|---|---|---|---|---|---|---|
| mine drainage water | SY18 | 152.5 | 82.58 | 53.7 | 212 | 150.1 | 279.8 | 7.67 | $Na^+ + Ca^{2+} + Mg^{2+}$—$HCO_3^- + Cl^-$ (26-A) |
| | SY19 | 79.98 | 75.6 | 37.41 | 78.2 | 76.26 | 218.7 | 7.66 | $Na^+ + Ca^{2+} + Mg^{2+}$—$HCO_3^-$ (5-A) |
| | SY20 | 44.14 | 47.75 | 15.96 | 22.59 | 31.03 | 250.9 | 7.6 | $Na^+ + Ca^{2+} + Mg^{2+}$—$HCO_3^- + SO_4^{2-}$ (12-A) |
| | SY21 | 34.3 | 102.5 | 30.17 | 34.6 | 204.5 | 205.4 | 7.34 | $Na^+ + Ca^{2+} + Mg^{2+}$—$HCO_3^-$ (9-A) |
| | SY22 | 80.99 | 23.87 | 14.48 | 14.77 | 52.45 | 263.7 | 7.35 | $Na^+ + Ca^{2+} + Mg^{2+}$—$HCO_3^- + SO_4^{2-}$ (12-A) |
| | SY23 | 39.93 | 41.78 | 19.31 | 29.41 | 66.5 | 187.3 | 7 | $Na^+ + Ca^{2+}$—$HCO_3^- + SO_4^{2-}$ (11-A) |
| | SY24 | 60.39 | 40.79 | 12.67 | 29.41 | 107 | 145 | 7.8 | $Na^+$—$HCO_3^-$ (7-A) |
| | SY25 | 58.76 | 51.73 | 23.84 | 45.18 | 79.88 | 257.3 | 7.75 | $Na^+ + Ca^{2+} + Mg^{2+}$—$HCO_3^-$ (5-A) |
| | SY26 | 62.99 | 34.82 | 26.55 | 15.64 | 64.34 | 295.9 | 7.75 | $Na^+ + Ca^{2+} + Mg^{2+}$—$HCO_3^-$ (5-A) |
| | SY27 | 63.11 | 68.64 | 38.02 | 22.05 | 227.7 | 237.8 | 7.8 | $Na^+ + Ca^{2+} + Mg^{2+}$—$HCO_3^- + SO_4^{2-}$ (12-A) |
| | SY28 | 60.83 | 25.86 | 21.72 | 6.48 | 67.53 | 224.5 | 8.3 | $Na^+ + Mg^{2+}$—$HCO_3^-$ (6-A) |

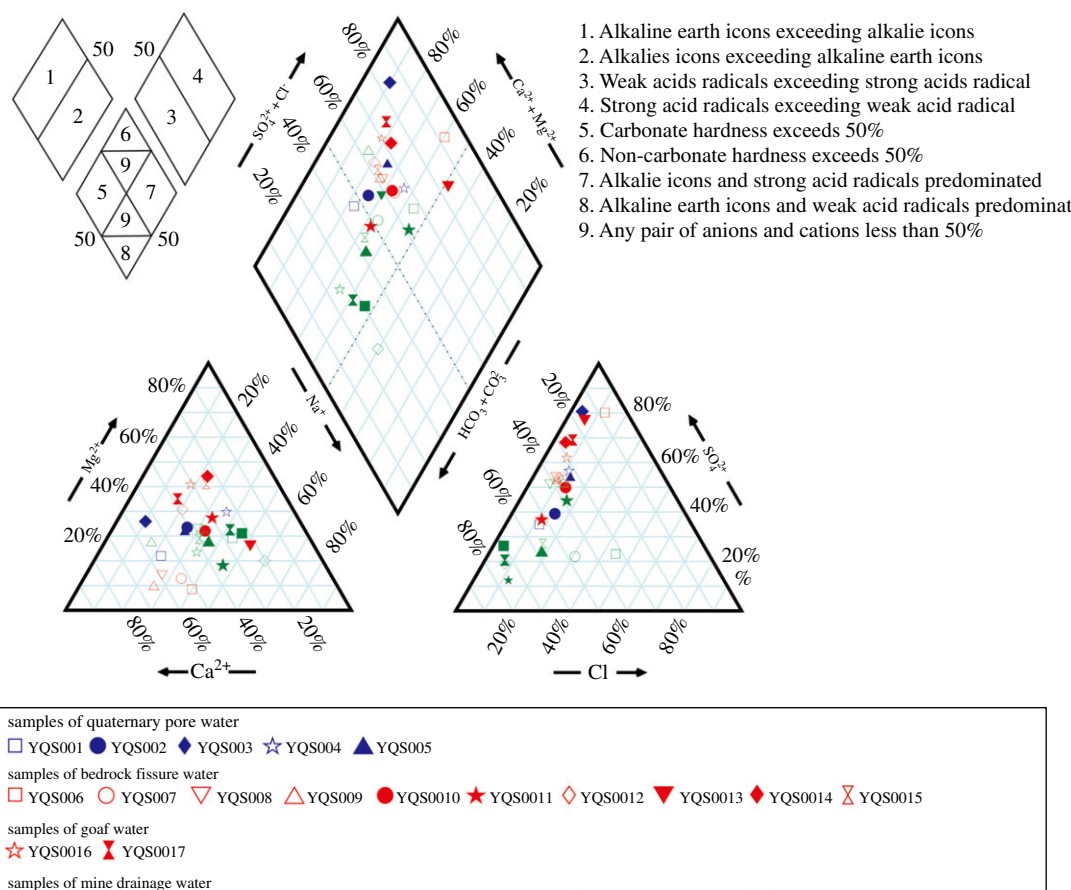

1. Alkaline earth icons exceeding alkalie icons
2. Alkalies icons exceeding alkaline earth icons
3. Weak acids radicals exceeding strong acids radical
4. Strong acid radicals exceeding weak acid radical
5. Carbonate hardness exceeds 50%
6. Non-carbonate hardness exceeds 50%
7. Alkalie icons and strong acid radicals predominated
8. Alkaline earth icons and weak acid radicals predominated
9. Any pair of anions and cations less than 50%

samples of quaternary pore water
□ YQS001  ● YQS002  ◆ YQS003  ☆ YQS004  ▲ YQS005

samples of bedrock fissure water
□ YQS006  ○ YQS007  ▽ YQS008  △ YQS009  ● YQS0010  ★ YQS0011  ◇ YQS0012  ▼ YQS0013  ◆ YQS0014  ⌧ YQS0015

samples of goaf water
☆ YQS0016  ⌶ YQS0017

samples of mine drainage water
□ YQS0018  ○ YQS0019  ☆ YQS0020  △ YQS0021  ⌧ YQS0022  ★ YQS0023  ◇ YQS0024  ▲ YQS0025  ⌶ YQS0026  ▽ YQS0027  ■ YQS0028

**Figure 11.** Piper diagrams of water samples in the study area.

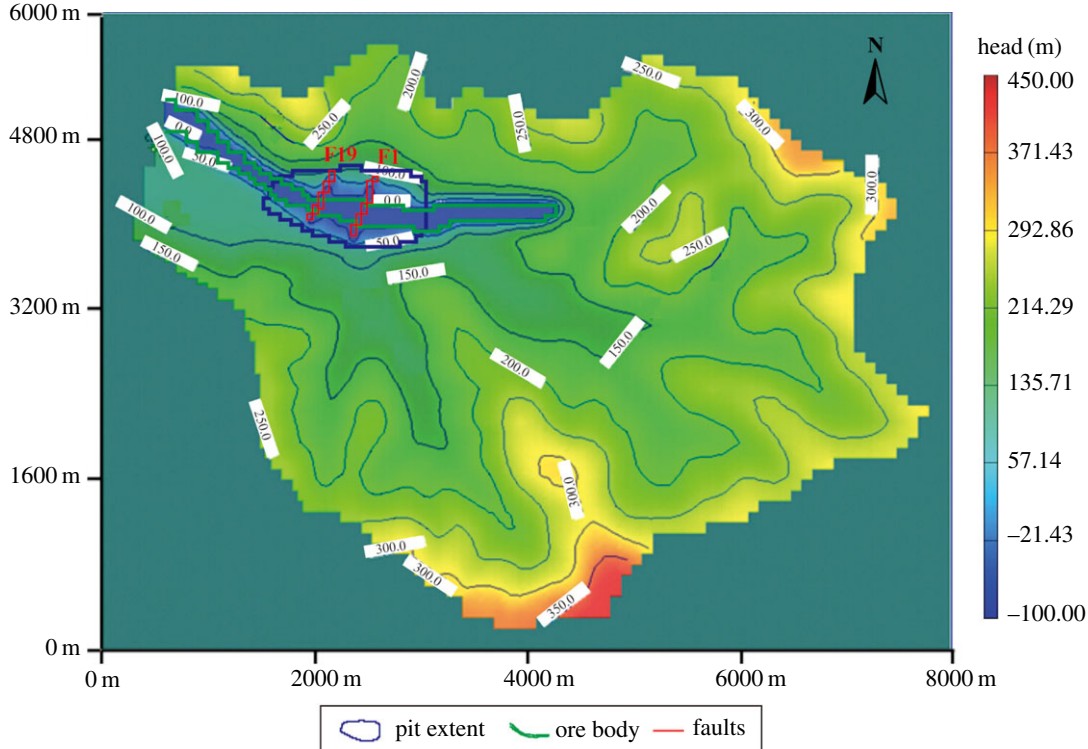

**Figure 12.** The groundwater head variation in layer 2.

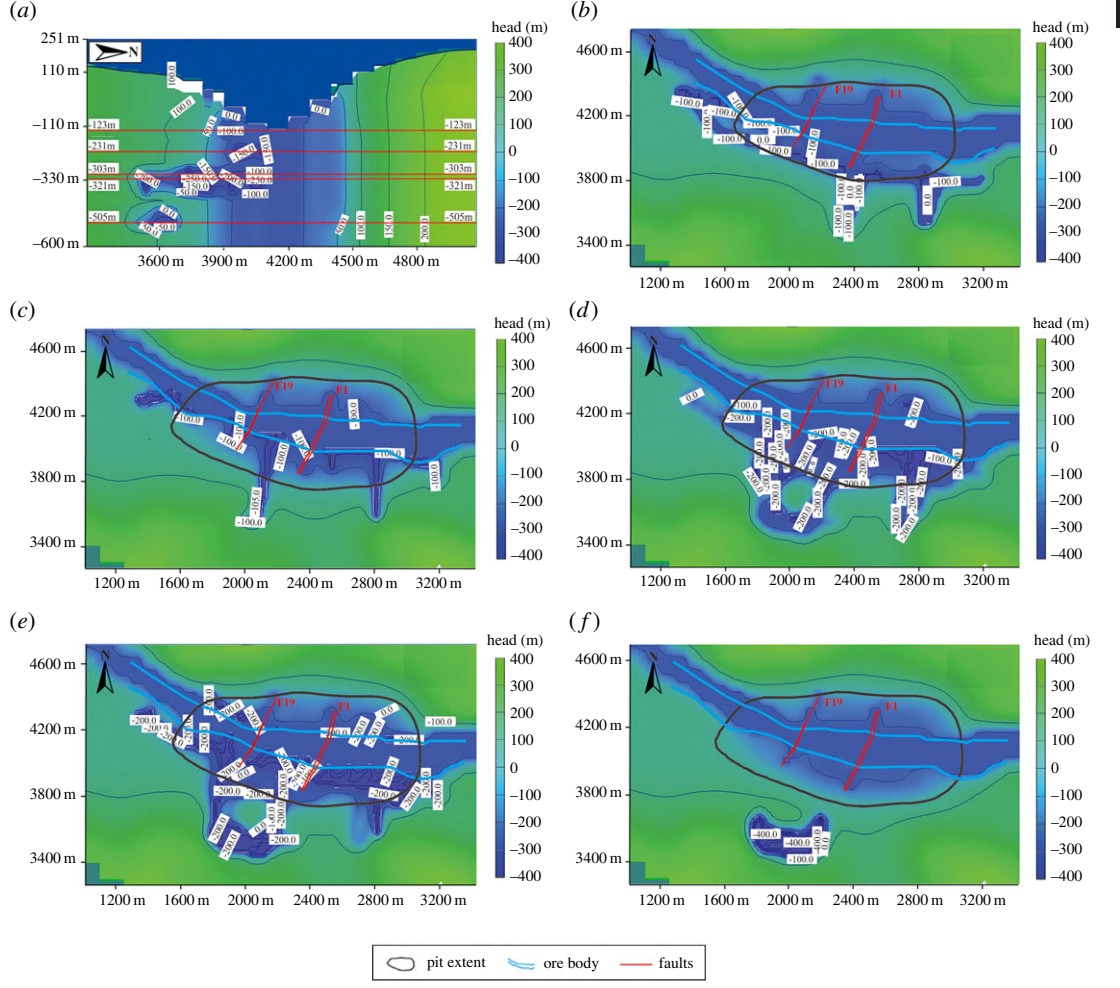

**Figure 13.** Groundwater head variation caused by roadway excavation. (*a*) Vertical cross-section, (*b*) −123 m horizontal cross-section, (*c*) −231 m horizontal cross-section, (*d*) −303 m horizontal cross-section, (*e*) −321 m horizontal cross-section and (*e*) −501 m horizontal cross-section

aquifer and in the faults, indicating that the faults (F19, F1) provide the seepage channel for the groundwater. As shown in figure 13, although the faults influence the groundwater head, the excavated roadways are the primary factor influencing groundwater migration. As shown in figure 13*a*, the source of water inrush in the roadways is mainly from the aquifer, while a small portion is from the aquitard. Groundwater flows into the mine roadways from all directions, indicating that the mine roadways influence groundwater migration in both the vertical and horizontal directions. The lower the elevation and the denser the arrangement of the roadways, the more the groundwater head changes.

MODPATH particle inverse tracking was used to determine the best path line of water tracer particles added to the mine roadways in the last time step of the model, with the direction of path line travelling from the recharge area into the discharge area [38]. As shown in figure 14, the direction of groundwater migration in the study area is generally from the aquitard to the aquifer. As shown in figure 14*b*,*f*, the direction of the path line is from the aquitard to the mine roadway. As shown in figure 14*c*−*e*, the direction of the path line is, again, from the aquifer to the mine roadway. Therefore, according to the simulation results, the source of water inrush at the mine roadway near the aquifer is the groundwater in the aquifer. However, the roadway near the aquitard provides a channel for groundwater migration—in other words, the source of the water inrush at the roadway near the aquitard is the groundwater in the aquitard.

## 4. Discussion

Mine water inrush has a great adverse impact on mining production, and induces disasters such as shaft submergence and underground debris flow, which lead to substantial economic losses and loss of life

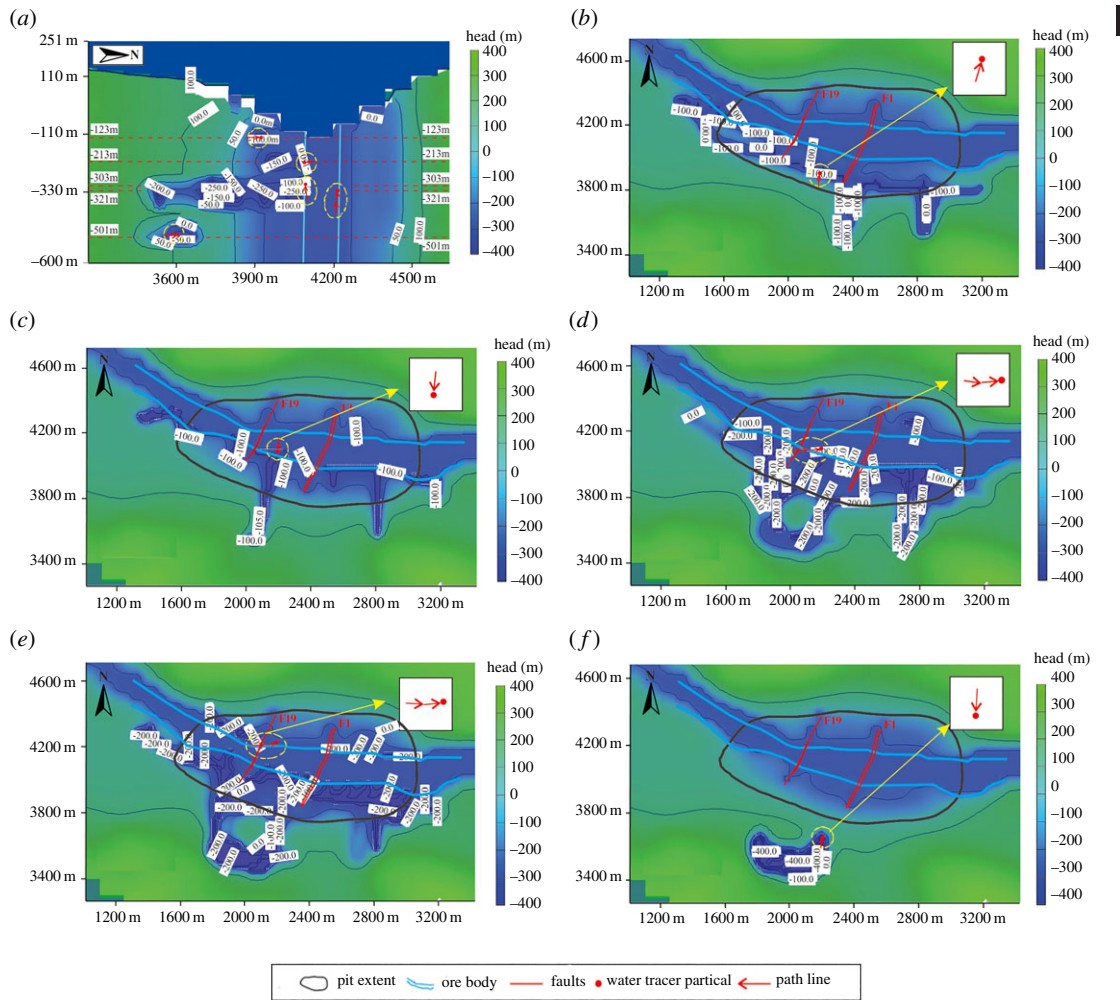

**Figure 14.** The path line of water tracer particles in the mine roadway. (*a*) Vertical cross-section, (*b*) −123 m horizontal cross-section, (*c*) −231 m horizontal cross-section, (*d*) −303 m horizontal cross-section, (*e*) −321 m horizontal cross-section and (*e*) −501 m horizontal cross-section.

[16,42]. In the process of transition from open-pit to underground mining, the large-scale open pit acts as a convergence point for surface water and groundwater. As mining depths increase and mining faces expand, the water accumulated in the pit flows into the mine roadway through the water seepage channels at the bottom of the pit and the fissures in the rock mass. Moreover, based on the above analysis, it is illustrated that the excavated roadway is the main factor affecting groundwater migration. Therefore, appropriate measures should be taken to prevent mine water inrush. Prevention measures can be divided into two aspects. (i) The first of these is surface prevention. The main source of groundwater in the study area is precipitation, and the main seepage channel is the outcropped fissures. Therefore, the work began with two aspects: reducing groundwater sources and reducing precipitation seepage channels. Precipitation was observed in the study area, and drainage projects were established to discharge the surface water in a timely manner. Additionally, a backfill layer of appropriate thickness was applied at the bottom of the pit to reduce the precipitation seepage channel. (ii) The second aspect is underground prevention. Based on the aforementioned results of the study, preventing water inrush at the mine roadway was primarily a matter of reducing the groundwater seepage channel. To address the faults and fissures around the mine roadway, a suitable plugging technology was adopted to effectively prevent the groundwater from flowing into the mine roadway. Simultaneously, a waterproof gate was installed in the mine roadway, and the water inflow was monitored.

The distribution of aquifer in the study area is nearly upright and the aquitard was on both sides of the aquifer to limit the groundwater flow in the aquifer, resulting in the poor drainage capacity in the study area. Moreover, based on the above analysis, it is illustrated that the water inflow in the mine roadways is large. Owing to the complex terrain and deep mining, traditional drainage methods

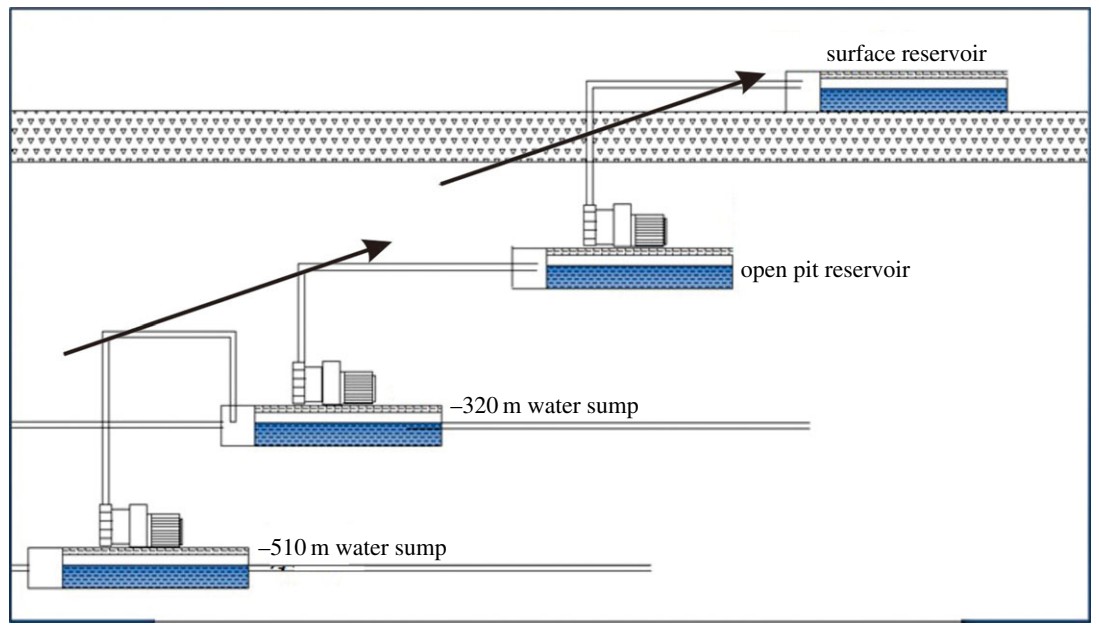

**Figure 15.** Multilevel drainage schematic diagram.

cannot rapidly and effectively solve the drainage problem in the study area [43,44]. Therefore, a multilevel drainage method was adopted for the treatment of mine water [45] (figure 15). (i) Water sumps and pumping stations were established at −320 and −501 m before underground mining. (ii) When mining above −320 m, the mine water was discharged into the water sump at the −320 m level via the drainage well and the drainage ditch. (iii) When mining below −320 m, the mine water was discharged into the water sump at the −501 m level via the drainage ditch. (iv) The mine water in the water sumps was discharged to the surface reservoir by the pumping station and the drainage pipe. (v) According to the results of the hydrochemical analysis in the study area, the turbidity, the concentration of Ferrum and Manganese and other indexes in the mine water exceed the prescribed range of industrial mill water [46]. Then, using the chemical coagulation and physical precipitation methods, the mine water was treated to meet the standard of mill water. (vi) The treated mine water was sent to a mineral processing plant through the drainage pipe.

## 5. Conclusion

This paper describes a comprehensive method, including hydrochemical analysis and numerical simulation, to study the characteristics of mine water inrush in the process of transition from open-pit to underground mining. The method proposed in this paper could visually reveal the migration law of groundwater and better solve the problem of the sources of water inrush. The results show that the direction of groundwater migration in the study area was generally from the aquitard to the aquifer. The source of mine water inrush was the groundwater in the aquifer around the mine roadway. A scientific basis for water inrush prevention during the transition from open-pit to underground mining was provided in this paper, and the comprehensive surface and underground prevention measures and multilevel drainage method were proposed to reduce water inrush in the mine roadway, which may serve as a useful reference for analogous engineering projects to solve similar water inrush problems.

Data accessibility. This article has no additional data.
Authors' contributions. H.Z. and B.Z. designed the study and wrote the manuscript. W.L. and Y.Y. finished the field investigation and collected the materials. N.X, L.S. and H.W. analysed the water inrush characteristics. All the authors gave their final approval for publication.
Competing interests. We declare we have no competing interests.
Funding. Financial support came from the National Natural Science Foundation of China (nos. 41572301, 61427802 and 41330634) and the Fundamental Research Funds for the Central Universities of China (nos. 2-9-2017-089 and 2-65-2018-108).
Acknowledgements. Additionally, the authors would like to acknowledge the editor and anonymous reviewers for their valuable comments and suggestions, which helped to significantly improve the quality of this paper.

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
