## [Reviewer comments · Royal Society Open Science]

Review History

RSOS-181402.R0 (Original submission)

Review form: Reviewer 1

Is the manuscript scientifically sound in its present form?

Yes

Are the interpretations and conclusions justified by the results?

Yes

Is the language acceptable?

Yes

Is it clear how to access all supporting data?

Yes

Do you have any ethical concerns with this paper?

No

Have you any concerns about statistical analyses in this paper?

I do not feel qualified to assess the statistics

Recommendation?

Major revision is needed (please make suggestions in comments)

Comments to the Author(s)

Dear Authors,

I send you the review of the manuscript RSOS-181402. In general, the manuscript is nearly conforming to the journal requirements, formats and quality. General remarks are focused on a better presentation of figures, and on including a comparison with previous works in a real discussion section. Some others compositional analysis of water could be included in order to get a complete view of the environmental concern authors are refereed to.

Section 2. Materials and methods

- Figure 1. Please include coordinates and a visible scale for the figure.
- I recommend authors to rename Section 2.1.1. as "Geological setting".
- Figures 2 and 3. Visible scales. Joint in only one figure could improve both figures. And include two representatives photographs: one for the open pit and another for the underground roadway.
- What about the foliation of metamorphic rocks?. Orientation?. Please draw with the correct orientation at the Figure 2.
- Faults are normal, inverse, slip-strike, ...? Displacement measured?
- 2.1.2. A reference for thickness of aquifer data is necessary.
- 2.2.1. P10 L7 "... (Fig. 5b)".
- A reference is necessary for SCM.
- An explanation (or reference) for water mineralisation groups is necessary.
- 2.2.2. What about the orientation of metamorphic foliation?. It could play an important role in the infiltration of groundwater.
- Figure 11. Include a scale.

Section 3. Result

- Rename as "Results".
- Inset values in Figures 13, 14 and 15 are so small. Please enlarge for a correct visualization.

Section 4. Discussion

- P23 L22. Authors talk about "... environmental pollution." but nothing is said about the potentially hazardous element contents in water composition. It could be very explicative to include metallic composition of water previous and after passing for the mine roadways.
- P23 L59. A multilevel drainage method is using for the treatment of water. In this sense, include the analytical results of the final chemical composition of water could be really interesting focused in an abroad audience.
- Finally, as said at the presentation paragraph of this review, a real Discussion section is necessary. Taken into account the presented data, a comparison with previous works on similar scenarios could significantly improved the manuscript and increase audience. There are several similar works for comparison.

I urge the authors not to view this solely as a critique but a chance to improve this paper to convey their ideas and data in the best possible and most accurate manner.

Warm regards

Review form: Reviewer 2

Is the manuscript scientifically sound in its present form?

Yes

Are the interpretations and conclusions justified by the results?

Yes

Is the language acceptable?

Yes

Is it clear how to access all supporting data?

Not Applicable

Do you have any ethical concerns with this paper?

No

Have you any concerns about statistical analyses in this paper?

No

Recommendation?

Major revision is needed (please make suggestions in comments)

Comments to the Author(s)

This paper is interesting because the mine water inrush is an important problem, which it must be addressed to ensure the water quality in mining areas. I believe that the article contributes to scientific-technical knowledge, is well-structured and addresses the objectives proposed.

However, in my opinion, there are several issues that should be corrected prior to publication. My suggestion is that the authors revise and reorder the text and include all the information necessary to improve some aspects for its publication.

Groundwater flow model

1. Justify the domain of the model
2. Explain the boundary conditions applied
3. Calibration process. It would be necessary to show the input parameters and the parameters obtained from the calibration; Level adjustment, Mass balance.
4. Justify the period time chosen to model

General

- The paper says that a methodology to prevent the mine water inrush has been implemented. Please, explain how this methodology has improved or not the mine water inrush. Make an assessment of the methodology used.
- The conclusion section is a repetition of what has been said in the introduction. Please, I recommend that you restructure this section and focus on the conclusions

Decision letter (RSOS-181402.R0)

09-Jan-2019

Dear Dr Zhang,

The editors assigned to your paper ("Water inrush characteristics and hazard effects during the

transition from open-pit to underground mining: a case study") have now received comments from reviewers. We would like you to revise your paper in accordance with the referee and Associate Editor suggestions which can be found below (not including confidential reports to the Editor). Please note this decision does not guarantee eventual acceptance.

Please submit a copy of your revised paper before 01-Feb-2019. Please note that the revision deadline will expire at 00.00am on this date. If we do not hear from you within this time then it will be assumed that the paper has been withdrawn. In exceptional circumstances, extensions may be possible if agreed with the Editorial Office in advance. We do not allow multiple rounds of revision so we urge you to make every effort to fully address all of the comments at this stage. If deemed necessary by the Editors, your manuscript will be sent back to one or more of the original reviewers for assessment. If the original reviewers are not available, we may invite new reviewers.

- Data accessibility

<http://datadryad.org/submit?journalID=RSOS&manu=RSOS-181402>

- Competing interests

- Authors' contributions

- Acknowledgements

- Funding statement

on behalf of Professor R. Kerry Rowe (Subject Editor)
openscience@royalsociety.org

Comments to Author:

Reviewers' Comments to Author:

Reviewer: 1

Comments to the Author(s)

Dear Authors,

I send you the review of the manuscript RSOS-181402. In general, the manuscript is nearly conforming to the journal requirements, formats and quality. General remarks are focused on a better presentation of figures, and on including a comparison with previous works in a real discussion section. Some others compositional analysis of water could be included in order to get a complete view of the environmental concern authors are refereed to.

Section 2. Materials and methods

- Figure 1. Please include coordinates and a visible scale for the figure.
- I recommend authors to rename Section 2.1.1. as "Geological setting".
- Figures 2 and 3. Visible scales. Joint in only one figure could improve both figures. And include two representative photographs: one for the open pit and another for the underground roadway.
- What about the foliation of metamorphic rocks?. Orientation?. Please draw with the correct orientation at the Figure 2.
- Faults are normal, inverse, slip-strike, ...? Displacement measured?
- 2.1.2. A reference for thickness of aquifer data is necessary.
- 2.2.1. P10 L7 "... (Fig. 5b)".
- A reference is necessary for SCM.
- An explanation (or reference) for water mineralisation groups is necessary.
- 2.2.2. What about the orientation of metamorphic foliation?. It could play an important role in the infiltration of groundwater.
- Figure 11. Include a scale.

Section 3. Result

- Rename as "Results".
- Inset values in Figures 13, 14 and 15 are so small. Please enlarge for a correct visualization.

Section 4. Discussion

- P23 L22. Authors talk about "... environmental pollution." but nothing is said about the potentially hazardous element contents in water composition. It could be very explicative to include metallic composition of water previous and after passing for the mine roadways.
- P23 L59. A multilevel drainage method is using for the treatment of water. In this sense, include the analytical results of the final chemical composition of water could be really interesting focused in an abroad audience.
- Finally, as said at the presentation paragraph of this review, a real Discussion section is necessary. Taken into account the presented data, a comparison with previous works on similar scenarios could significantly improved the manuscript and increase audience. There are several similar works for comparison.

I urge the authors not to view this solely as a critique but a chance to improve this paper to convey their ideas and data in the best possible and most accurate manner.

Warm regards

Reviewer: 2

Comments to the Author(s)

This paper is interesting because the mine water inrush is an important problem, which it must be addressed to ensure the water quality in mining areas. I believe that the article contributes to scientific-technical knowledge, is well-structured and addresses the objectives proposed. However, in my opinion, there are several issues that should be corrected prior to publication. My suggestion is that the authors revise and reorder the text and include all the information necessary to improve some aspects for its publication.

Groundwater flow model

1. Justify the domain of the model
2. Explain the boundary conditions applied
3. Calibration process. It would be necessary to show the input parameters and the parameters obtained from the calibration; Level adjustment, Mass balance.

4. Justify the period time chosen to model

General

- The paper say that a methodology to prevent the mine water inrush has been implemented. Please, explain how this methodology has improved or not the mine water inrush. Make an assessment of the methodology used.
- The conclusion section is a repetition of what has been said in the introduction. Please, I recommend that you restructure this section and focus on the conclusions

Author's Response to Decision Letter for (RSOS-181402.R0)

See Appendix A.

RSOS-181402.R1 (Revision)

Review form: Reviewer 1

Is the manuscript scientifically sound in its present form?

Yes

Are the interpretations and conclusions justified by the results?

Yes

Is the language acceptable?

Yes

Is it clear how to access all supporting data?

Yes

Do you have any ethical concerns with this paper?

No

Have you any concerns about statistical analyses in this paper?

No

Recommendation?

Accept as is

Comments to the Author(s)

All suggestions and comments have been taken into account and, therefore, the manuscript is ready for publishing.

Regards,

Review form: Reviewer 2

Is the manuscript scientifically sound in its present form?

Yes

Are the interpretations and conclusions justified by the results?

Yes

Is the language acceptable?

Yes

Is it clear how to access all supporting data?

No

Do you have any ethical concerns with this paper?

No

Have you any concerns about statistical analyses in this paper?

No

Recommendation?

Accept as is

Comments to the Author(s)

The authors have addressed the majority of my previous comments and I think that the authors have done a good job in answering to all the questions raised during the review process and the result is a much improved manuscript. So I feel that the manuscript can be now accepted for publication. Thank you.

Decision letter (RSOS-181402.R1)

12-Feb-2019

Dear Dr Zhang,

I am pleased to inform you that your manuscript entitled "Water inrush characteristics and hazard effects during the transition from open-pit to underground mining: a case study" is now accepted for publication in Royal Society Open Science.

on behalf of Prof R. Kerry Rowe (Subject Editor)
openscience@royalsociety.org

Reviewer comments to Author:

Reviewer: 1

Comments to the Author(s)

All suggestions and comments have been taken into account and, therefore, the manuscript is ready for publishing.

Regards,

Reviewer: 2

Comments to the Author(s)

The authors have addressed the majority of my previous comments and I think that the authors have done a good job in answering to all the questions raised during the review process and the result is a much improved manuscript. So I feel that the manuscript can be now accepted for publication. Thank you.

Follow Royal Society Publishing on Twitter: [@RSocPublishing](https://twitter.com/RSocPublishing)

Reviewer 1

Thank you very much for your review, as well as your valuable suggestions on this paper. Your comments regarding inappropriate presentations were extremely valuable and helpful in revising and improving the paper. Thus, we have studied the comments carefully and have made the appropriate corrections. We redraw Fig. 1, Fig. 2, Fig. 10, Fig. 12, Fig. 13 and Fig. 14 in the the revised manuscript. Questions and simple replies were summarized in the table below. And detail information and modification were list item by item after the table. All inappropriate presentation has been modified and updated, and all amendments have been highlighted in red in the revised manuscript.

Questions	Simple replies
1. Figure 1. Please include coordinates and a visible scale for the figure.	Thanks for your suggestions. We have redrawn Fig. 1 of the revised manuscript.
2. I recommend authors to rename Section 2.1.1. as "Geological setting".	Thanks for your suggestions. We have renamed Section 2.1.1 in Line 58 .
3. Figures 2 and 3. Visible scales. Joint in only one figure could improve both figures. And include two representatives photographs: one for the open pit and another for the underground roadway.	Thank you for your question. We have jointed them in only one figure as shown in Fig. 2 of the revised manuscript.
4. What about the foliation of metamorphic rocks? Orientation? Please draw with the correct orientation at the Figure 2.	Thank you for your question. In the revised manuscript, we added the description of the foliation in Line 72-74 . And we have drawn the foliation of phyllite with the correct orientation in Fig. 2(b) of the revised manuscript.
5. Faults are normal, inverse, slip-strike, ...? Displacement measured?	Thank you for your question. In the revised manuscript, we added the description of displacement, dip and dip angles of faults in Line 76-80 .

6. A reference for thickness of aquifer data is necessary.	Thank you for your question. In the revised manuscript, we added the description of the thickness of aquifer data in Line 99-100 .
7. P10 L7 "... (Fig. 5b)"	Thank you for your question. In the revised manuscript, we reordered the pictures in the paper.
8. A reference is necessary for SCM.	Thank you for your question. In the revised manuscript, we added the description of the Schukalev classification method in Line 114-118 .
9. An explanation (or reference) for water mineralisation groups is necessary.	Thank you for your question. In the revised manuscript, we added the description of the water mineralization in Line 118-122 .
10. What about the orientation of metamorphic foliation?. It could play an important role in the infiltration of groundwater.	Thank you for your question. We have given a clear explanation on the influence of the orientation of metamorphic foliation in reply to Q10 and added the description of the foliation in Line 72-74 .
11. Figure 11. Include a scale.	Thank you for your question, we have added a plotting scale in Fig. 10 of the revised manuscript.
12. Rename as "Results".	Thanks for your suggestions. We have renamed Section 3 in Line 208 .
13. Inset values in Figures 13, 14 and 15 are so small. Please enlarge for a correct visualization.	Thanks for your suggestions. We have enlarged the inset values for a correct visualization in Fig. 12, 13 and 14 of the revised manuscript. Additionally, we have modified the legends in Fig. 13 and 14 of the revised manuscript.
14. P23 L22. Authors talk about "... environmental pollution." but nothing is said about the potentially hazardous element contents in water composition. It could be very explicative to include metallic	Thanks for your suggestions. We have given a clear explanation on environmental pollution in reply to Q14. In order to make the theme of this article clearer, we made some changes in Line 267-273 .

composition of water previous and after passing for the mine roadways.	
15. A multilevel drainage method is using for the treatment of water. In this sense, include the analytical results of the final chemical composition of water could be really interesting focused in an abroad audience.	Thank you for your comments. We have given a clear explanation on the final chemical composition in reply to Q15. In the revised manuscript, we added explanation on the description of the multilevel drainage method in Line 284-297 .
16. Finally, as said at the presentation paragraph of this review, a real Discussion section is necessary. Taken into account the presented data, a comparison with previous works on similar scenarios could significantly improved the manuscript and increase audience. There are several similar works for comparison.	Thanks for your suggestions. In order to make the theme of this article clearer, we reconstructed the structure of the section of discussion and added some references in Line 267-297 .

The problems you have suggested:

1. Figure 1. Please include coordinates and a visible scale for the figure.

Reply:

Thank you for your valuable suggestions. Indeed, as you stated, we neglected coordinates of the study area. We have fully absorbed your suggestions and referred to the geological data in the study area. Then, We modified the figure of the geographic location of the study area in the revised manuscript, including coordinates and a visible scale (as shown in Fig. 1 of the revised manuscript).

Fig. 1 Geographic location of the study area

2. I recommend authors to rename Section 2.1.1. as "Geological setting".

Reply:
 Thanks for your suggestions. We have fully absorbed your suggestions and renamed Section 2.1.1. Line 58 "2.1.1 Geological setting".

3. Figures 2 and 3. Visible scales. Joint in only one figure could improve both figures. And include two representatives photographs: one for the open pit and another for the underground roadway.

Reply:
 Thank you for your valuable suggestions. Indeed, as you stated, Fig. 2 and 3 barely have a concise and accurate description of the geological conditions of the study area in the previous manuscript. We have fully absorbed your suggestions and jointed them in only one figure (as shown in Fig. 2 of the revised manuscript). In order to reflect the location and field condition of the open pit and mine roadway, we referred to the field data and photographs and added two representatives photographs in Fig.3 of previous manuscript (the Fig. 2(b) of the revised manuscript) : one for the open pit and another for the mine roadway.

Fig.2 Geological condition of the study area

4. What about the foliation of metamorphic rocks?. Orientation?. Please draw with the correct orientation at the Figure 2.

Reply:

Thank you for your question. Unfortunately, we have not been able to give a clear explanation on the foliation in the previous manuscript. Indeed, the foliation is important for describing the characteristic of metamorphic rocks. Moreover, the orientation of metamorphic rocks has a great influence on the hydrogeological condition of the study area. Therefore, we referred to the geological data of the study area and described the foliation of the metamorphic rocks in detail. We have fully absorbed your suggestions and added the description of the foliation of the phyllite in the revised manuscript in **Line 72-74** “The phyllite mainly contains two sets of dominant foliations, both sets of

them were closed before excavation. The dip angles are 10° and 80° respectively (as shown in Fig. 2 of the previous manuscript and Fig. 2(b) of the revised manuscript).”

5. Faults are normal, inverse, slip-strike, ...? Displacement measured?

Reply:

Thank you for your question. Unfortunately, we didn't give a clear explanation on the faults in the previous manuscript. Indeed, the faults provide the seepage channel for the groundwater and influence on the groundwater distribution in the study area. As mining depths increase and mining faces expand, the faults have more influence on the groundwater head. Therefore, it is necessary to refer to the geological data of the study area and describe the fault in detail. We have fully absorbed your suggestions and added the description of displacement, dip and dip angles of faults in the revised manuscript in Line 76-80 “F13 is normal fault with the vertical displacement of 160m, which occur in the north-east direction and whose dip angle is approximately $67-90^\circ$. F20 is strike-slip fault with the horizontal displacement is 70m, which occur in the north-east direction and whose dip angle is approximately 45° . F1 and F19 are strike-slip faults, with the horizontal displacement are 40-130m and 30-100m respectively, which occur in the south-east direction and whose dip angles are approximately 74° and $83-90^\circ$ respectively.”

6. A reference for thickness of aquifer data is necessary.

Reply:

Thank you for your question. Indeed, as you stated, the thickness of aquifer has great influence on the water inrush problem. The main aquifers in the study area are Quaternary overburden and magnetite-quartzite (Fig. 2b). The Quaternary overburden is a phreatic aquifer with good water-richness and the magnetite-quartzite is the bedrock fissure aquifer cut by multiple faults. Moreover, the distribution of aquifer in the study area is nearly upright and the aquitard on both sides of the aquifer to limit the groundwater flow in the aquifer. Therefore, as mining depths increase and mining faces expand, the water inrush problem in the study area is more serious. we referred to the hydrogeological data of the study area and added the description of the thickness of aquifer data in

the revised manuscript in Line 99-100 “The magnetite-quartzite is the bedrock fissure aquifer, 55-195 m thick, nearly upright, cut by multiple faults.”

7. P10 L7 "... (Fig. 5b)"

Reply:
Thank you for your question. In the revised manuscript, we reorder the pictures in the paper, as shown in Fig. 4(b) of the previous manuscript (as shown in Fig. 4b of the revised manuscript).

8. A reference is necessary for SCM.

Reply:
Thank you for your question. As you pointed out, some references are necessary for explanation on the Shukalev classification method (SCM). In order to analyse the hydrogeological connectivity, the hydrochemical analysis method was proposed to analyse the hydrochemical characteristics in the study area. Hydrochemical classification is a comprehensive indicator for analysing the hydrochemical characteristics of groundwater. At present, the SCM is one of the most widely used methods in hydrochemical type analysis. The results of the hydrochemical analysis of water samples from the study area was placed in the classification table according to their chemical composition. Based on the similar chemical composition between different water samples, the migration law of groundwater was analysed. We have fully absorbed your suggestions and referred to related references. The description of the Schukalev classification method was added in the revised manuscript in Line 114-118 “Artificial exploitation significantly changes the concentrations of major ions in the groundwater, resulting in the complex and diverse hydrochemical characteristics in the mining area [32]. Hydrochemical classification is a comprehensive indicator for analysing the hydrochemical characteristics of groundwater. At present, the Shukalev classification method (SCM) is the most widely used in hydrochemical classification analysis, reflecting the migration law of groundwater [33].”

[33] Liu Q, Sun YJ, Xu ZM, Xu G. 2018 Application of the comprehensive identification model in analyzing the source of water inrush. Arab. J. Geosci. **11**, 189. (doi: 10.1007/s12517-018-3550-2)

9. An explanation (or reference) for water mineralisation groups is necessary.

Reply:

Thank you for your question. Unfortunately, we have not been able to give a clear explanation on the definition of mineralization of water in the previous manuscript. Mineralization of water refers to the sum of bicarbonates, chlorides, sulfates, nitrates, and various metals carbonates such as calcium, magnesium, aluminum, and manganese in the water. The SCM is based on the main ions concentrations and mineralization in groundwater. Water samples cannot be classified in detail based solely on ion concentration. Ion concentrations greater than 25 meq% are categorized into 49 types. The mineralization of water is categorized into four types: Type A (<1.5 g/L), Type B (1.5-10 g/L), Type C (10-40 g/L), and Type D (>40 g/L). The advantage of this classification is that it is easy to understand and can be used to organize water analysis data. We have fully absorbed your suggestions and added the description of the SCM in the revised manuscript in Line 118-122 “The SCM is based on the main ions concentrations and mineralization in groundwater. Ion concentrations greater than 25 meq% are categorized into 49 types (Table 1). The mineralization of water is categorized into four types: Type A (<1.5 g/L), Type B (1.5-10 g/L), Type C (10-40 g/L), and Type D (>40 g/L). The two classifications—ion content and water mineralization—are combined into a designation of the form "1-A", for example, where 1 denotes the ion concentration and A denotes the water mineralization [34, 35].”

10. What about the orientation of metamorphic foliation?. It could play an important role in the infiltration of groundwater.

Reply:

Thank you for your question. Indeed, as you said the orientation of metamorphic foliation plays an important role in the infiltration of groundwater. The well-developed foliation provide provide the seepage channel for the groundwater. However, through related reference and the hydrogeological data of the study area, we found that both sets of the foliation of phyllite were closed before excavation. Therefore, the foliation of phyllite was barely influenced on the groundwater migration.

We have fully absorbed your suggestions and added the description of the foliation of the phyllite in the revised manuscript in Line 72-74 “The phyllite mainly contains two sets of dominant foliations, both sets of them were closed before excavation. The dip angles are 10° and 80° respectively.”

11. Figure 11. Include a scale.

Reply:
Thank you for your question. Indeed, as you stated, we neglected coordinates in Fig. 11 of the previous manuscript. We have fully absorbed your suggestions and referred to the geological data in the study area. We modified the figure of the layout of the mine roadways in the study area, including a plotting scale in the Fig. 10 of the revised manuscript.

Fig. 10 Layout of the mine roadways in the study area

12. Rename as "Results".

Reply:

Thanks for your suggestions. We have fully absorbed your suggestions and renamed Section 3 in Line 208 "3.Results."

13. Inset values in Figures 13, 14 and 15 are so small. Please enlarge for a correct visualization.

Reply:

Thanks for your suggestions. Indeed, small values could not fully reflect the information in the previous figures. We have fully absorbed your suggestions and enlarge the inset values for a correct visualization in Fig. 13, 14 and 15 of the previous manuscript (as shown in Fig. 12, 13 and 14 of the revised manuscript). In addition, in order to accurately reflect the information in Fig.14 and 15, we modified the legends in Fig. 13 and 14 of the revised manuscript.

Fig. 12 The groundwater head variation in Layer

Fig. 13 Groundwater head variation caused by roadway excavation

Fig. 14 The path line of water tracer particles in the mine roadway

14. P23 L22. Authors talk about "... environmental pollution." but nothing is said about the potentially hazardous element contents in water composition. It could be very explicative to include metallic composition of water previous and after passing for the mine roadways.

Reply:
 Thanks for your suggestions. Unfortunately, we have not been able to give a clear explanation in the previous manuscript. Indeed, the environmental pollution is a problem in the process of the

excavation of the mine roadway. However, this paper focus on the study of the characteristics of mine water inrush in the process of transition from open-pit to underground mining. A comprehensive method that incorporates hydrochemical analysis and numerical simulation was proposed to reveal the migration law of groundwater and analyse the sources of water inrush. Provide a scientific basis for water inrush prevention in mining districts. The environmental pollution caused by metallic composition of water previous and after passing for the mine roadways was not considered, but this can be considered as a direction for future research. In order to make the theme of this article clearer, we made some changed in Line 267-273 “Mine water inrush has a great adverse impact on mining production, and induces disasters such as shaft submergence and underground debris flow, which lead to substantial economic losses and loss of life [42, 43]. In the process of transition from open-pit to underground mining, the large-scale open pit acts as a convergence point for surface water and groundwater. As mining depths increase and mining faces expand, the water accumulated in the pit flew into the mine roadway through the water seepage channels at the bottom of the pit and the fissures in the rock mass. Moreover, based on the above analysis, it is illustrated that the excavated roadway is the main factor affecting groundwater migration. Therefore, appropriate measures should be taken to prevent mine water inrush.”

15. P23 L59. A multilevel drainage method is using for the treatment of water. In this sense, include the analytical results of the final chemical composition of water could be really interesting focused in an abroad audience.

Reply:

Thank you for your valuable suggestion. In the last manuscript, we did not give a clear explanation on the multilevel drainage method, and we were sorry for the confusion to you. Aiming to rapidly and effectively solve the drainage problem and prevent further water inrush problem in the study area, a multilevel drainage method was adopted for the large water inflow in the mine roadways. The distribution of aquifer in the study area is nearly upright and the aquitards on both the sides of aquifer to limit the groundwater flow in the aquifer, resulting in the poor drainage capacity in the study area. Moreover, based on the analysis in the paper, it is illustrated that the water inflow in the

mine roadways is large. Due to the complex terrain and deep mining, a multilevel drainage method was proposed. In addition, in order to comprehensively utilize mine drainage, the mine water was treated and supplied to the mineral processing plant. According to the results of the hydrochemical analysis in the study area, the turbidity, the concentration of Ferrum and Manganese and other indexes in the mine water exceed the prescribed range of industrial mill water. Then, we referred to the reuse of urban recycling water-water quality standard for industrial uses (Table1). Using chemical coagulation method and physical precipitation method, the mine water was treated to meet the standard of mill water.

Table1 Urban recycling water-water quality standard for industrial uses

Serial number	Control project	Industrial water
1	PH \leq	6.5-8.5
2	Turbidity (NTU) \leq	5
3	Chromaticity \leq	30
4	BOD5 (mg/L) \leq	10
5	CODcr (mg/L) \leq	60
6	Ferrum (mg/L) \leq	0.3
7	Manganese (mg/L) \leq	0.1
8	Chlorine (mg/L) \leq	250
9	Silicon dioxide \leq	30
10	Total hardness (CaCO ₃) (mg/L) \leq	450
11	Total alkalinity (CaCO ₃) (mg/L) \leq	350
12	Sulphate (mg/L) \leq	250
13	Ammonia nitrogen (N) (mg/L) \leq	10
14	Total phosphorus (P) (mg/L) \leq	1
15	Total dissolved solids (mg/L) \leq	1000
16	Fecal coliform (/L) \leq	2000
17	Petroleum (mg/L) \leq	1
18	Anionic surfactant (mg/L) \leq	0.5

In the revised manuscript, we give a clear explanation on the multilevel drainage method in Line 284-297 “The distribution of aquifer in the study area is nearly upright and the aquitards on both sides of the aquifer to limit the groundwater flow in the aquifer, resulting in the poor drainage capacity in the study area. Moreover, based on the above analysis, it is illustrated that the water inflow in the mine roadways is large. Due to the complex terrain and deep mining, traditional drainage methods cannot rapidly and effectively solve the drainage problem in the study area. [44, 45]. Therefore, a multilevel drainage method was adopted for the treatment of mine water [46] (Fig. 15). (1) Water sumps and pumping stations were established at -320 m and -501 m before underground mining. (2) When mining above -320 m, the mine water was discharged into the water sump at the -320 m level via the drainage well and the drainage ditch. (3) When mining below -320 m, the mine water was discharged into the water sump at the -501 m level via the drainage ditch. (4) The mine water in the water sumps was discharged to the surface reservoir by the pumping station and the drainage pipe. (5) According to the results of the hydrochemical analysis in the study area, the turbidity, the concentration of Ferrum and Manganese and other indexes in the mine water exceed the prescribed range of industrial mill water [47]. Then, using chemical coagulation method and physical precipitation method, the mine water was treated to meet the standard of mill water. (6) The treated mine water was sent to a mineral processing plant through the drainage pipe.”

[44] Wu Q, Fan ZL, Zhang ZW, Zhou, WF. 2014 Evaluation and zoning of groundwater hazards in Pingshuo No. 1 underground coal mine, Shanxi Province, China. *Hydrogeol. J.* **22**, 1693-1705. (doi: 10.1007/s10040-014-1138-9)

[45] Li HJ, Chen QT, Shu ZY, Li L, Zhang YC. 2018 On prevention and mechanism of bed separation water inrush for thick coal seams: a case study in China. *Environ. Earth Sci.* **77**, 12. (doi: 10.1007/s12665-018-7952-y)

[47] GB/T 19923-2005, 2005. The reuse of urban recycling water-water quality standard for industrial uses. Ministry of Housing and Urban-Rural Development of the People's Republic of China, Beijing (in Chinese).

16. Finally, as said at the presentation paragraph of this review, a real Discussion section is necessary. Taken into account the presented data, a comparison with previous works on similar scenarios could significantly improved the manuscript and increase audience. There are several similar works for comparison.

Reply:

Thanks for your suggestions. Indeed, as your statement, a real discussion in this paper is necessary. Referring to the relevant literature, combined with the conclusions of this paper, the discussion section was revised. In fact, there are have similar engineering projects, such as the Chah-Gaz iron mine in Iran, Udachny mine in Russia and Shirengou iron mine in China. Existing studies have focused on optimizing the transition from open-pit to underground mining. However, as mining depths increase and mining faces expand, more serious water inrush problems can occur during the transition from open-pit to underground mining. Therefore, the prevention and treatment of mine water inrush were proposed in this paper, which will serve as a valuable reference for analogous engineering cases.

The distribution of aquifer in the study area nearly is upright and the aquitards on both the sides of aquifer to limit the groundwater flow in the aquifer, resulting in the poor drainage capacity in the study area. In addition, the large-scale open pit acts as a convergence point for surface water and groundwater. As mining depths increase and mining faces expand, the water accumulated in the pit flew into the mine roadway through the water seepage channels at the bottom of the pit and the cracks in the rock mass. Moreover, based on the results in this paper, it is illustrated that the water inflow in the mine roadways is large. Therefore, the surface and underground comprehensive prevention measures and multilevel drainage method were proposed to reduce water inrush in the mine roadway. The proposed methods have successfully adopted in the Yanqianshan iron mine located in Liaoning province, China and may serve as a useful reference for analogous engineering projects to solve similar water inrush problems.

In order to make the theme of this article clearer, we reconstructed the structure of the section of

discussion in Line 267-297 “Mine water inrush has a great adverse impact on mining production, and induces disasters such as shaft submergence and underground debris flow, which lead to substantial economic losses and loss of life [42, 43]. In the process of transition from open-pit to underground mining, the large-scale open pit acts as a convergence point for surface water and groundwater. As mining depths increase and mining faces expand, the water accumulated in the pit flow into the mine roadway through the water seepage channels at the bottom of the pit and the fissures in the rock mass. Moreover, based on the above analysis, it is illustrated that the excavated roadway is the main factor affecting groundwater migration. Therefore, appropriate measures should be taken to prevent mine water inrush. Prevention measures can be divided into two aspects. (1) The first of these is surface prevention. The main source of groundwater in the study area is precipitation, and the main seepage channel is the outcropped fissures. Therefore, the work began with two aspects: reducing groundwater sources and reducing precipitation seepage channels. Precipitation was observed in the study area, and drainage projects were established to discharge the surface water in a timely manner. Additionally, a backfill layer of appropriate thickness was applied at the bottom of the pit to reduce the precipitation seepage channel. (2) The second aspect is underground prevention. Based on the aforementioned results of the study, preventing water inrush at the mine roadway was primarily a matter of reducing the groundwater seepage channel. To address the faults and fissures around the mine roadway, a suitable plugging technology was adopted to effectively prevent the groundwater from flowing into the mine roadway. Simultaneously, a waterproof gate was installed in the mine roadway, and the water inflow was monitored.

The distribution of aquifer in the study area nearly is upright and the aquitards on both sides of the aquifer to limit the groundwater flow in the aquifer, resulting in the poor drainage capacity in the study area. Moreover, based on the above analysis, it is illustrated that the water inflow in the mine roadways is large. Due to the complex terrain and deep mining, traditional drainage methods cannot rapidly and effectively solve the drainage problem in the study area. [44, 45]. Therefore, a multilevel drainage method was adopted for the treatment of mine water [46] (Fig. 15). (1) Water sumps and pumping stations were established at -320 m and -501 m before underground mining. (2) When mining above -320 m, the mine water was discharged into the water sump at the -320 m level via the

drainage well and the drainage ditch. (3) When mining below -320 m, the mine water was discharged into the water sump at the -501 m level via the drainage ditch. (4) The mine water in the water sumps was discharged to the surface reservoir by the pumping station and the drainage pipe. (5) According to the results of the hydrochemical analysis in the study area, the turbidity, the concentration of Ferrum and Manganese and other indexes in the mine water exceed the prescribed range of industrial mill water [47]. Then, using chemical coagulation method and physical precipitation method, the mine water was treated to meet the standard of mill water. (6) The treated mine water was sent to a mineral processing plant through the drainage pipe.

Reviewer 2

Thank you very much for your review, as well as your valuable suggestions on this paper. Your comments regarding inappropriate presentations were extremely valuable and helpful in revising and improving the paper. Thus, we have studied the comments carefully and have made the appropriate corrections. We modified the legend in Fig. 9 of the revised manuscript. Questions and simple replies were summarized in the table below. And detail information and modification were list item by item after the table. All inappropriate presentation has been modified and updated, and all amendments have been highlighted in red in the revised manuscript.

Questions	Simple replies
1. Justify the domain of the model	Thanks for your comments. We have added the description of the domain of the model in the revised manuscript in Line 153-155 .
2. Explain the boundary conditions applied	Thanks for your comments. We have added the description of the boundary condition and initial condition respectively in Line 140-145 and in Line169-174 .
3. Calibration process. It would be necessary to show	Thanks for your comments. We have made a clearer

the input parameters and the parameters obtained from the calibration; Level adjustment, Mass balance.	explanation on the calibration in the revised manuscript in Line192-198. In addition, we modified the legend in Fig. 9 of the revised manuscript.
4. Justify the period time chosen to model	Thanks for your comments. We have made a clearer explanation on the period time chosen to model in the revised manuscript in Line160-163.
5. The paper say that a methodology to prevent the mine water inrush has been implemented. Please, explain how this methodology has improved or not the mine water inrush. Make an assessment of the methodology used.	Thanks for your comments. We have made major revisions to the description of the methodology and the purpose of the study in the revised manuscript. Some related modification have been done in the revised manuscript as below: Line 8-17, Line 37-39, Line 48-55 and Line 107-110.
6. The conclusion section is a repetition of what has been said in the introduction. Please, I recommend that you restructure this section and focus on the conclusions	Thanks for your comments. In order to make the theme of this article clearer, we reconstructed the section of the conclusion in Line 301-309.

The problems you have suggested:

1. Justify the domain of the model

Reply:

Thanks for your comments. Unfortunately, we have not been able to give a clear explanation in the previous manuscript. Indeed, the domain chosen to the model has great influence on the simulation of the groundwater seepage field. The study area is surrounded by mountains with an elevation of 210-386 m to the south, east and north, while the area to the west consists of plains with an average elevation of approximately 93 m. Due to the excavation of the mine pit, the permeability of rock mass in the mine pit is increase with the outcrop area failure, the well-developed fissures, and the

influence of the weathering. Therefore, in order to eliminate the uncertainty of boundary conditions of fractured rock mass, the model domain should be extended to the boundary of the hydrogeological unit. Since the groundwater on either side of the watershed does not flow to the other side, the catchment area of the study area was chosen as the domain of groundwater seepage model. We have fully absorbed your suggestions and added the description of the domain of the model in the revised manuscript in Line 153-155 “In addition to the engineering area, the model domain should be extended to the boundary of the hydrogeological unit to eliminate the uncertainty of boundary conditions of fractured rock mass. Therefore, the catchment area of the study area was chosen as the domain of groundwater seepage model.”

2. Explain the boundary conditions applied

Reply:

Thanks for your comments. Unfortunately, we have not been able to give a clear explanation in the previous manuscript. Indeed, as you stated, we have neglected a detailed description in the text of the boundary conditions in the previous manuscript. The boundary condition reflects the groundwater relationship between the study area and its surrounding environment. Through the determined boundary condition and initial condition, the particular solution of calculate partial differential equation of groundwater seepage was obtained, which meet the actual groundwater seepage field in the study area. The domain of the groundwater seepage model is the catchment area of the study area, bounded by the watershed. Since the groundwater on either side of the watershed does not flow to the other side, the volume of water at the watershed is zero, the model boundary was set as the no flow in the second type of boundary condition. Referring to the geological data in the study area, it is illustrated that the main drainage channel in the study area is the Guyu River, which located to the south of the mine pit. Therefore, the boundary of the river was set as fixed water level. Due to the model focus on the impact on groundwater migration and drainage effect of mine roadways, the boundaries of mine roadway was set as the boundary of fixed water level.

We have fully absorbed your suggestions and added the description of the boundary condition and

initial condition respectively in Line 140-145 “The solution of the governing partial differential equation satisfies the second type of boundary condition expressed by Eq. (2). The boundary of fixed water level is expressed by Eq. (3) [37]:

$$K_{\bar{n}} \left. \frac{\partial h}{\partial \bar{n}} \right|_{\Gamma_2} = q(x, y, z, t) \quad x, y, z \in \Gamma_2, t \geq 0 \quad (2)$$

$$\nabla \cdot (\rho \bar{u}) = Q \quad (3)$$

where ρ is the density of fluid (kg/m^3); q is the volume of water laterally flowing into or out of the aquifer per unit area and per unit time under the second type of boundary.” and in Line 169-174 “According to the observation wells data in the study area, the initial water head of the study area was obtained and assigned to the unsteady flow model. The domain of the groundwater seepage model was the catchment area of the study area, bounded by the watershed. The groundwater on either side of the watershed does not flow to the other side, the volume of water at the watershed is zero. Therefore, the boundary condition of model was set as the no flow in the second type of boundary condition. The boundaries of the river and mine roadway were set as the boundary of fixed water level (Fig. 6 and Fig. 10).”

3. Calibration process. It would be necessary to show the input parameters and the parameters obtained from the calibration; Level adjustment, Mass balance.

Reply:

Thanks for your comments. Unfortunately, we have not been able to give a clear explanation in the previous manuscript. Indeed, as you stated, a clear explanation on the calibration process of the model should be given to ensure that it has the capacity to simulate the actual groundwater seepage field in the study area. Due to the lack of field data, we selected the monitoring data in 2012 to calibrate the model.

Based on the monitoring data, we calibrated the numerical model by adjusting the seepage parameters and boundary conditions. Obtained from previous studies and field investigation, the permeability coefficient partition used in the model, Region I was the phreatic aquifer, Region IV was the bedrock fissure aquifer and Regions II and III were the aquitard. Permeability coefficient of

the aquitard is $1e^{-10}$ cm/s in the study area. However, due to the excavation of the mine pit, the permeability of aquitard in the mine pit increased with the outcrop area failure, the well-developed fissures, and the influence of the weathering. Therefore, in the process of calibration, the permeability coefficient of the Regions II was reasonably improved to $1e^{-6}$ cm/s to better reflect the actual situation.

We have fully absorbed your suggestions and make a clearer explanation on the calibration in the revised manuscript in Line192-198 “The model should be calibrated to ensure that it has the capacity to simulate the actual groundwater seepage field in the study area [40]. Based on the monitoring data, the numerical model was calibrated by adjusting the seepage parameters and boundary conditions. Comparing simulated values and measured values of the groundwater head, the capacity of simulating the actual groundwater seepage field in the study area can be evaluated. The results are shown in Fig. 9. The difference between the simulated values and the measured values was less than 1 m, and the confidence level was greater than 95%. The model calibration results prove that the three-dimensional groundwater seepage model effectively reflected the actual groundwater seepage field of the study area.” In addition, in order to accurately reflect the information in Fig.10 of the previous manuscript, we modified the legend in Fig. 9 of the revised manuscript.

Fig. 9 Comparison between calculated head and observed head

4. Justify the period time chosen to model

Reply:

Thanks for your comments. Unfortunately, we have not been able to give a clear explanation in the previous manuscript. Changes in the groundwater head after roadway excavation were simulated in the calibrated model. After five years of calculation, the groundwater head change was basically stable. Therefore, the period time chosen to model is a time step of 30 days for a period of 60 months (January 2008-December 2012). We have fully absorbed your suggestions and made a clearer explanation on the period time chosen to model in the revised manuscript in **Line160-163** “Calculations were performed for unsteady flow with a time step of 30 days for a period of 60 months (January 2008-December 2012). After five years of calculation, the groundwater head change was basically stable, indicating that the period time chosen to model is reasonable.”

5. The paper say that a methodology to prevent the mine water inrush has been implemented. Please, explain how this methodology has improved or not the mine water inrush. Make an assessment of the methodology used.

Reply:

Thanks for your comments. In the previous manuscript, we did not show the meaning of our research very well, and we were sorry for the confusion to you. In fact, there are have similar engineering projects, such as the Chah-Gaz iron mine in Iran, Udachny mine in Russia and Shirengou iron mine in China. Existing studies have focused on optimizing the transition from open-pit to underground mining. However, as mining depths increase and mining faces expand, more serious water inrush problems can occur during the transition from open-pit to underground mining.

The distribution of aquifer in the study area is nearly upright and the aquitards on both sides of the aquifer to limit the groundwater flow in the aquifer, resulting in the poor drainage capacity in the study area. In the process of transition from open-pit to underground mining, the large-scale open pit

acts as a convergence point for surface water and groundwater. As mining depths increase and mining faces expand, the water inrush problem is more serious. This paper describes a comprehensive method, including hydrochemical analysis and numerical simulation, to study the characteristics of mine water inrush in the process of transition from open-pit to underground mining. Moreover, based on the results of the study, the appropriate methods for prevention and treatment of mine water inrush were proposed. The proposed methods have successfully adopted in the Yanqianshan iron mine located in Liaoning province, China and may serve as a useful reference for analogous engineering projects to solve similar water inrush problems.

We have fully absorbed your valuable suggestions and made major revisions to the description of methodology and the purpose of the study in the revised manuscript. Some related modifications have been done in revised manuscript as below.

Line 8-17 “During the transition from open-pit to underground mining in iron ore mines, water inrush is a prominent problem for mine safety and production. In this paper, a comprehensive method that incorporates hydrochemical analysis and numerical simulation is proposed to analyse the characteristics of water inrush during the transition from open-pit to underground mining. The proposed method revealed the hydrologic connectivity of groundwater and analyse the source of mine water inrush in the Yanqianshan iron mine located in Liaoning province, China. The results show that the excavated mine roadway is the primary factor affecting groundwater migration and that the source of the mine water inrush is the groundwater in the aquifer around the mine roadway. Moreover, based on the results of the study, appropriate methods for prevention and treatment of mine water inrush were proposed. This approach provides a novel idea for the assessment of water inrush hazards and will serve as a valuable reference for analogous engineering cases.”

Line 37-39 “In order to provide a scientific basis for water inrush prevention in mining districts, it is of vital importance to determine the cause and source of water inrush.”

Line 48-55 “In this paper, a comprehensive method incorporating hydrochemical analysis and numerical simulation was proposed to analyse the characteristics of water inrush during the transition from open-pit to underground mining. The hydrochemical characteristics of water samples from the study area were analysed. Through the Schukalev classification method and Piper diagrams, the hydraulic connection of groundwater in the study area was revealed. A three-dimensional groundwater seepage model was built to analyse the characteristics of mine water inrush. With the use of MODPATH particle inverse tracking, the migration law of groundwater was analysed and the source of water inrush at the mine roadway was identified. Based on this analysis, prevention and treatment measures were proposed to solve the problem of mine water inrush.

Line 107-110 “Aiming to describe a comprehensive study of the characteristics of water inrush during the process of transition from open-pit to underground mining, a comprehensive method incorporating hydrochemical analysis and numerical simulation is proposed. Based on the aforementioned studies, prevention and treatment measures were proposed to solve the problem of mine water inrush. The corresponding flowchart is shown in Fig. 5.

6. The conclusion section is a repetition of what has been said in the introduction. Please, I recommend that you restructure this section and focus on the conclusions

Reply:

Thanks for your comments. As you stated, the previous manuscript indeed had some problems with the conclusion section. The poor conclusion section seriously affected the readability of the previous manuscript. We have fully absorbed your suggestions, referred to the relevant literature, and combined with the results of this paper, the conclusion section was revised. The revised conclusion section of the manuscript is in Line 301-309 “This paper describes a comprehensive method, including hydrochemical analysis and numerical simulation, to study the characteristics of mine water inrush in the process of transition from open-pit to underground mining. The method proposed in this paper could visually revealed the migration law of groundwater and better solved the problem

of the sources of water inrush. The results show that the direction of groundwater migration in the study area was generally from the aquitard to the aquifer. The source of mine water inrush was the groundwater in the aquifer around the mine roadway. A scientific basis for water inrush prevention in during transition from open-pit to underground mining was provided in this paper, and the surface and underground comprehensive prevention measures and multilevel drainage method were proposed to reduce water inrush in the mine roadway, which may serve as a useful reference for analogous engineering projects to solve the similar water inrush problems.”

Additional modifications

1. In order to have a detailed description in the text of the numerical model, We have added the description of faults, the sources of water input, the sources of water output and observation wells in the revised manuscript in the Line 163-168 “According to the hydrogeological data, the hydraulic conductivity of the faults of F1 and F19 is strong, influencing on the groundwater distribution in the mining area. Therefore, only the influence of the faults of F1 and F19 on the groundwater seepage field was considered in the model. The sources of water input to the groundwater system is recharge from precipitation. The sources of water output from the groundwater system are groundwater evaporation and the Guyu River. The head observation wells were set in the model to observe the simulated values of the groundwater head, as shown in Fig. 6.”

2. We have carefully checked the revised manuscript and found some mistakes in the figures. There are some grammatical errors in Fig.6 of the previous manuscript. We modified the figure of the flowchart of analyses conducted during this study (as shown in the Fig.5 of the revised manuscript). In addition, the water samples are not fully displayed in Fig.12 of the previous manuscript. We modified the figure of the piper diagrams of water samples in the study area (as shown in the Fig.11 of the revised manuscript).

Fig. 5 Flowchart of analyses conducted during this study

Fig. 11 Piper diagrams of water samples in the study area